# The molecular infrastructure of glutamatergic synapses in the mammalian forebrain

Julia Peukes[1†], Charlie Lovatt[2‡], Conny Leistner[2§], Jerome Boulanger[1#], Dustin R Morado[1], Martin JG Fuller[2], Wanda Kukulski[1¶], Fei Zhu[3], Noboru H Komiyama[3,4], John AG Briggs[1**], Seth GN Grant[3,4], René AW Frank[1,2*]

[1]MRC Laboratory of Molecular Biology, Francis Crick Avenue, Cambridge, United Kingdom; [2]Astbury Centre for Structural Molecular Biology, School of Biomedical Sciences, Faculty of Biological Sciences, University of Leeds, Leeds, United Kingdom; [3]Genes to Cognition Program, Institute for Neuroscience and Cardiovascular Research, University of Edinburgh, Edinburgh, United Kingdom; [4]Simons Initiative for the Developing Brain, Centre for Discovery Brain Sciences, University of Edinburgh, Edinburgh, United Kingdom

**\*For correspondence:**
R.Frank@leeds.ac.uk

**Present address:** [†]Chan Zuckerberg Imaging Institute, Redwood City, United States; [‡]Rosalind Franklin Institute, Didcot, United Kingdom; [§]Harvard Medical School, Boston, United States; [#]Science for Life Laboratory, Department of Biochemistry and Biophysics, Stockholm University, Stockholm, Sweden; [¶]Institute of Biochemistry and Molecular Medicine, University of Bern, Bern, Switzerland; [**]Max Planck Institute of Biochemistry, Martinsried, Germany

**Competing interest:** The authors declare that no competing interests exist.

## eLife Assessment

Peukes et al. report **compelling** ultrastructures of excitatory synapses in the mouse forebrain that will serve as a reference for future work in the field. Their **important** findings using correlated fluorescence and cryo-electron tomography challenge the textbook view of synaptic structure that emerged from chemically fixed and metal-stained tissues. Instead of a post-synaptic density, these authors reveal the architecture of the cytoskeletal, neurotransmitter receptor clusters, and organelles in the 'synaptoplasm'.

**Abstract** Glutamatergic synapses form the vast majority of connections within neuronal circuits, but how these subcellular structures are molecularly organized within the mammalian brain is poorly understood. Conventional electron microscopy using chemically fixed, metal-stained tissue has identified a proteinaceous, membrane-associated thickening called the 'postsynaptic density' (PSD). Here, we combined mouse genetics and cryo-electron tomography to determine the 3D molecular architecture of fresh isolated and anatomically intact synapses in the adult forebrain. The native glutamatergic synapse did not consistently show a higher density of proteins at the post-synaptic membrane, thought to be characteristic of the PSD. Instead, a 'synaptoplasm' consisting of cytoskeletal elements, macromolecular complexes, and membrane-bound organelles extended throughout the pre- and post-synaptic compartments. Snapshots of active processes gave insights into membrane remodeling processes. Clusters of up to 60 ionotropic glutamate receptors were positioned inside and outside the synaptic cleft. Together, these information-rich tomographic maps present a detailed molecular framework for the coordinated activity of synapses in the adult mammalian brain.

## Introduction

Glutamate is the major chemical neurotransmitter in the mammalian brain that mediates communication within neuronal circuits at specialized cell-cell junctions called synapses. The molecular

composition of glutamatergic synapses is highly complex, reflecting the many different molecular machines required to subserve diverse synaptic functions, including learning and memory (*Chua et al., 2010*; *Ebrahimi and Okabe, 2014*; *Frank et al., 2016*; *Nakahata and Yasuda, 2018*). Indeed, more than a thousand different synaptic proteins have been identified in the mouse and human forebrain (*Bayés et al., 2011*; *Roy et al., 2018*). How this ensemble of macromolecules is structurally organized within adult mammalian brain synapses is currently unknown.

Cryo-electron microscopy (cryoEM) has the potential to reveal the native 3-dimensional molecular architecture of mammalian brain synapses, but with the limitation that samples must be vitreous and less than ~300 nm thick. Although primary neurons grown on EM grids can overcome this constraint (*Asano et al., 2015*; *Tao et al., 2018*), the environment in which in vitro cultured cells grow differs markedly from tissues. Indeed, primary neurons in vitro appear unable to reach the molecular and functional maturity of neurons from the brain itself (*Frank et al., 2016*; *Harris and Pettit, 2007*). While brain synapses can be isolated by biochemical fractionation (*Whittaker, 1959*), these enrichment steps delay cryopreservation by at least an hour and involve non-physiological buffers that together cause deterioration of certain macromolecular structures. An example is the cytoskeleton, particularly microtubules, which rapidly depolymerize within a matter of 1–3 minutes if the provision of nucleotide triphosphates or cellular integrity becomes compromised (*Mitchison, 1995*; *Pollard and Mooseker, 1981*).

To minimize deterioration of tissue samples, we developed an 'ultra-fresh' sample preparation workflow without fractionation by combining mouse genetic labelling of synapses (*Zhu et al., 2018*) with cryogenic correlated light and electron microscopy (cryoCLEM) and cryo-electron tomography (cryoET) (*Figure 1A*). Guided by the mouse genetic fluorescent label in our cryoCLEM workflow, we also reconstructed in-tissue cryoET volumes of synapses from vitreous cryo-sections of mouse cortex and hippocampus. These ultra-fresh and in-tissue structural data indicate that the near-physiological molecular architecture of glutamatergic synapses was highly variable in composition, copy number, and distribution of organelles and macromolecular assemblies. The molecular density of protein complexes in the postsynaptic cytoplasm was variable and did not indicate a conserved higher density of proteins at the postsynaptic membrane corresponding to a postsynaptic density (PSD).

## Results

### CryoCLEM of adult brain glutamatergic synapses

To determine the molecular architecture of adult forebrain excitatory synapses, we used knockin mice, in which the gene encoding Psd95 (*Dlg4*) was engineered to include an in-frame, C-terminal fusion of the fluorescent tag EGFP (*Dlg4*$^{GFP/GFP}$) (*Broadhead et al., 2016*; *Frank et al., 2016*; *Zhu et al., 2018*). Psd95 is a membrane-associated cytoplasmic protein that concentrates exclusively within mature glutamatergic synapses (*Chen et al., 2000*; *El Husseini et al., 2000*) and forms supercomplexes with ionotropic glutamate receptors that reside in the postsynaptic membrane (*Frank et al., 2016*). Fresh *Dlg4*$^{GFP/GFP}$ mouse forebrains, encompassing cortex and hippocampus, were homogenized in ice-cold ACSF, then immediately vitrified on EM grids. Sample preparation was completed within 2 min, the time taken to cull the mouse, dissect and homogenize the forebrain, and plunge-freeze the sample on a blotted cryoEM grid (*Figure 1A*).

The GFP signal from *Dlg4*$^{GFP/GFP}$ knockin mice pinpoints the location of mature glutamatergic synapses without over-expression or any detectable effect on synapse function (*Broadhead et al., 2016*). Thus, cryoEM grids were imaged on a cryogenic fluorescence microscope (cryoFM), revealing Psd95-GFP puncta (*Figure 1B*), absent in WT control samples (*Figure 1—figure supplement 1A*), which resembled those observed by confocal fluorescent imaging of the brain (*Zhu et al., 2018*). Using a cryoCLEM workflow, the same grids were then imaged by cryoEM, and fluorescent Psd95-GFP puncta were mapped onto the cryoEM image (*Figure 1B*). These cryoCLEM images showed Psd95-GFP puncta that were invariably associated with membrane-bound subcellular compartments (100–1300 nm diameter) (*Figure 1B*).

To resolve the molecular architecture within Psd95-containing puncta, we acquired 93 Psd95-GFP cryoCLEM-guided cryo-electron tomographic tilt series from five adult mouse forebrains. Aligned tilt series were reconstructed to produce three-dimensional tomographic maps as shown in *Figure 1C and D* (see also *Figure 1—source data 1* and *Figure 1—videos 1–7* showing representative tomographic

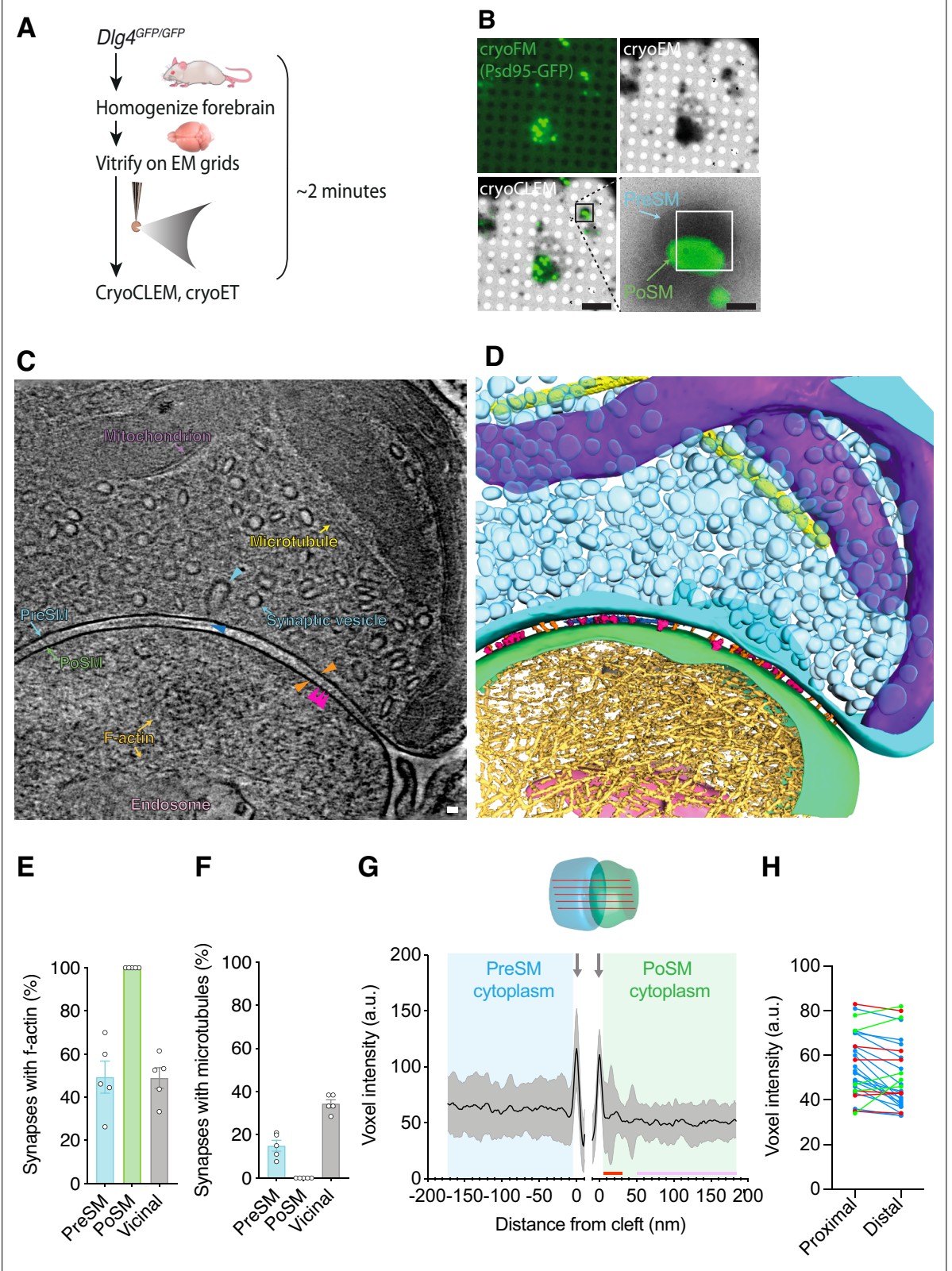

**Figure 1.** Cryogenic correlated light and electron microscopy (CryoCLEM)-targeted cryo-electron tomography (cryoET) of mouse forebrain synapses. (**A**) Schematic of the 'ultra-fresh' preparation of synapses for cryoCLEM. *Dlg4*$^{GFP/GFP}$ knockin mice were culled, forebrain was dissected, homogenized in ice-cold artificial cerebrospinal fluid, and plunge-frozen on cryoEM grids. 2 min was the time taken to cull, dissect, and cryopreserve samples on EM grids. (**B**) CryoCLEM of a cryoEM grid square containing an ultra-fresh synapse preparation. *Top left*, cryoFM image of a holey carbon grid square. *Top*

*Figure 1 continued*

*right*, cryoEM image of the same grid square shown in top left. *Bottom left*, merged image of cryoFM and cryoEM images indicating location of Psd95-GFP puncta. Scale bar, 5 μm. Black box indicates region enlarged in *Bottom right*, showing Psd95-GFP associated with the PoSM. White rectangle indicates region where images for the tomogram shown in C were acquired. Scale bar, 500 nm. (C) Tomographic slice of Psd95-containing glutamatergic synapse. The PoSM (green) was identified by Psd95-GFP cryoCLEM and the PreSM (cyan) was identified by its tethering to the PoSM and the prevalence of synaptic vesicles. Salient organelle and macromolecular constituents are indicated: Purple arrow, mitochondrion. Yellow arrow, microtubule. Cyan arrow, synaptic vesicle, and cyan arrowhead indicating intermediate of vesicle fission/fusion. Gold arrow, F-actin filament. Pink arrow, putative endosomal compartment. Orange arrowheads, transsynaptic macromolecular complex bridging PreSM and PoSM. Magenta arrowheads, postsynaptic membrane proteins with extracellular domains extending 13–14 nm into the synaptic cleft. Blue arrow, lateral matrix of macromolecules in synaptic cleft. Scale bar, 20 nm. The tomographic map shown here is representative of 93 tomograms obtained across 5 mouse forebrain preparations. (D) 3D segmentation of membranes and macromolecules in a representative tomographic volume of a Psd95-containing glutamatergic synapse. Coloured as in C. (E) Prevalence of branched filamentous actin networks in presynaptic membrane compartments (PreSM, cyan), Psd95-containing postsynaptic membrane compartments (PoSM, green), and neighbouring non-synaptic membrane compartments (Vicinal, grey). Data points are per mouse, for 5 adult *Dlg4*$^{GFP/GFP}$ mouse forebrain samples. Error bars, SEM. (F) Same as E but for microtubules. (G) Molecular crowding of the PreSM and PoSM cytoplasm in ultra-fresh synapse preparation. *Top*, schematic showing the measurement of molecular density estimated with multiple line profiles plots of voxel intensity within each synapse tomogram. *Bottom*, voxel intensity (a.u., arbitrary units) profile plots of presynaptic (cyan) and postsynaptic (green) cytoplasm. Profiles were aligned to the lipid membrane peak of the PreSM and PoSM (grey arrows). The average intensity profile of 27 synapses from 3 mice is shown in black with 1 standard deviation in grey. Red and purple bar, proximal and distal regions of PoSM cytoplasm, respectively, as analysed in H. (H) Comparison of average molecular density profile in regions 5–30 nm (proximal) and 50–200 nm (distal) from the PoSM of each synapse. Regions proximal to the PoSM correspond to locations in which a PSD is a conserved feature in conventional EM. Blue, red and green datapoints, synapses with cytoplasm proximal to the PoSM that contained significantly higher, lower, and not significantly different density from distal regions, respectively (two-tailed *t* test with Bonferroni correction, p<0.05, n=682 and 5673 voxels).

The online version of this article includes the following video, source data, and figure supplement(s) for figure 1:

**Source data 1.** A spreadsheet detailing the organelle and macromolecular constituents within each tomogram of the ultra-fresh synapse cryoET dataset, including within the PreSM and PoSM of Psd95-GFP containing synapses and additional subcellular compartments in the vicinity.

**Figure supplement 1.** Cryogenic fluorescence microscope (CryoFM), docked vesicles and actin in Psd95-GFP-containing synapses.

**Figure supplement 2.** Molecular density analysis within synapse tomographic volumes.

**Figure supplement 3.** Tomographic slices and molecular density profiles of ultra-fresh Psd95-GFP synapses.

**Figure supplement 4.** Tomographic slices and molecular density profiles of ultra-fresh Psd95-GFP synapses.

**Figure supplement 5.** Tomographic slices and molecular density profiles of ultra-fresh Psd95-GFP synapses.

**Figure supplement 6.** Tomographic slices and molecular density profiles of ultra-fresh Psd95-GFP synapses.

**Figure supplement 7.** Tomographic slices and molecular density profiles of ultra-fresh Psd95-GFP synapses.

**Figure supplement 8.** Tomographic slices and molecular density profiles of ultra-fresh Psd95-GFP synapses.

**Figure supplement 9.** Tomographic slices and molecular density profiles of ultra-fresh Psd95-GFP synapses.

**Figure 1—video 1.** Reconstructed tomographic volumes of Psd95-GFP-containing ultra-fresh synapses.
https://elifesciences.org/articles/100335/figures#fig1video1

**Figure 1—video 2.** Reconstructed tomographic volumes of Psd95-GFP-containing ultra-fresh synapses.
https://elifesciences.org/articles/100335/figures#fig1video2

**Figure 1—video 3.** Reconstructed tomographic volumes of Psd95-GFP-containing ultra-fresh synapses.
https://elifesciences.org/articles/100335/figures#fig1video3

**Figure 1—video 4.** Reconstructed tomographic volumes of Psd95-GFP-containing ultra-fresh synapses.
https://elifesciences.org/articles/100335/figures#fig1video4

**Figure 1—video 5.** Reconstructed tomographic volumes of Psd95-GFP-containing ultra-fresh synapses.
https://elifesciences.org/articles/100335/figures#fig1video5

**Figure 1—video 6.** Reconstructed tomographic volumes of Psd95-GFP-containing ultra-fresh synapses.
https://elifesciences.org/articles/100335/figures#fig1video6

**Figure 1—video 7.** Reconstructed tomographic volumes of Psd95-GFP-containing ultra-fresh synapses.
https://elifesciences.org/articles/100335/figures#fig1video7

volumes of ultra-fresh synapses). As expected, all GFP-positive membranes were attached to a neighbouring membrane, of which 78% were closed compartments and enriched in synaptic vesicles that characterize the presynaptic compartment. All but three of the presynaptic compartments (96%) contained docked synaptic vesicles tethered 2–8 nm from the presynaptic membrane (*Figure 1—figure supplement 1B*). Thus, our Psd95-GFP cryoCLEM workflow enabled identification of the

pre- and post-synaptic membranes of glutamatergic synapses (hereon referred to as PreSM and PoSM, respectively).

The high quality of the tomographic maps enabled assignment of cytoskeletal elements within the synapse, including extensive networks of filamentous actin (F-actin). These were identifiable with a diameter of 7 nm and apparent helical arrangement of subunits (*Hanson, 1967*) in all PoSM compartments (*Okamoto et al., 2007*; *Figure 1C-E*, *Figure 1—figure supplement 1C*) and in 49 ± 7% (mean ± SEM, n=5 mice) of PreSM compartments (*Figure 1E* and *Figure 1—figure supplement 1D*). In PreSM compartments that lacked an apparent F-actin cytoskeletal network, we cannot exclude the possibility that actin is present in monomeric form or very short actin filaments, which cannot be unambiguously identified in our tomograms. Microtubules were identified (25 nm diameter) in 15 ± 2% of PreSM compartments (*Figure 1C, D and F*), but as expected, were absent from the PoSM. Notwithstanding the mechanical stress of detaching synaptic terminals to cryopreserve whole synapses, this extensive actin and microtubular cytoskeleton corroborates the maintenance of macromolecular structures near to their physiological context.

## Molecular crowding within the synapse

We first assessed the overall architecture of macromolecules within the synapse by measuring macromolecular crowding (*Smith and Langmore, 1992*) (see methods and *Figure 1—figure supplement 2* detailing how molecular density was measured and how confounding errors were avoided). Based on conventional EM experiments of fixed, dehydrated, heavy-metal-stained tissue, glutamatergic synapses are marked by the presence of a 30–50 nm layer of densely packed proteins opposite the active zone on the cytoplasmic side of the PoSM, called the postsynaptic density (PSD) that defines every glutamatergic synapse (*Bourne and Harris, 2008*; *Gray, 1959*; *Sheng and Kim, 2011*). However, within fresh, adult brain synapse structures reported here, this region of higher density at the postsynaptic membrane was not a conserved feature of all Psd95-GFP-containing glutamatergic synapses (*Figure 1g* and see also the central density profiles of each synapse in *Figure 1—figure supplements 3–9*). The distribution of molecular densities was variable with half the population of synapses showing a significant relative increase in molecular density in proximal versus distal regions (5–30 nm versus 50–200 nm from the PoSM) of cytoplasm in the postsynaptic compartment (*Figure 1H*). The remaining subpopulations of synapses did not have a higher density or contained a significantly lower relative molecular density at regions proximal to the PoSM (*Figure 1H*, two-tailed *t* test with Bonferroni correction, p<0.05, n=682 and 5673 voxels). These data suggest that a higher density of proteins characteristic of the PSD is not a conserved feature with adult mouse brain glutamatergic synapses (*Tao et al., 2018*).

To examine further molecular crowding within anatomically intact postsynaptic compartments, fresh, acute brain tissues of 7 adult *Dlg4*$^{GFP/GFP}$ mice were cryopreserved by high-pressure freezing. Next, 70–150 nm thin, vitreous cryo-sections of the cortex and hippocampus were collected by cryo-ultramicrotomy (*Figure 2A*; *Zuber et al., 2005*). Glutamatergic synapses were identified by cryoCLEM (*Figure 2B*) and cryo-electron tomograms were collected revealing the native in-tissue 3D architecture of glutamatergic synapses (*Figure 2C-D*, *Figure 2—figure supplement 1A* and *Figure 2—videos 1–2*). The PoSM was identified by cryoCLEM and the PreSM by the presence of numerous synaptic vesicles. These in-tissue 3D tomographic maps also resolved individual cytoskeletal elements, membrane proteins, and numerous other subcellular structures that were consistent with those observed in ultra-fresh synapse cryoET data (*Figure 2—source data 1*).

Molecular density profiles of in-tissue synapse cryoET data showed that the density of proteins in the cytoplasm proximal to the PoSM relative to distal regions was highly variable (*Figure 2E* and see also profiles of each synapse in *Figure 2—figure supplements 1–6*). Less than half the population of synapses contained a significant relative increase in molecular density in regions of the cytoplasm proximal to the PoSM (5–30 nm versus 50–200 nm from the PoSM, *Figure 2F*). The remaining subpopulations of synapses lacked a higher density or contained a significantly lower relative molecular density at regions proximal to the PoSM (*Figure 2F*, two-tailed *t*-test with Bonferroni correction, p<0.05, n=682, and 5673 voxels). No significant difference was observed comparing synapses from the cortex versus hippocampus, as expected (*Zhu et al., 2018*). The variability of molecular crowding in these near-physiological in-tissue synapses was consistent with the ultra-fresh synapse tomograms, indicating that the presence of even a slight increase

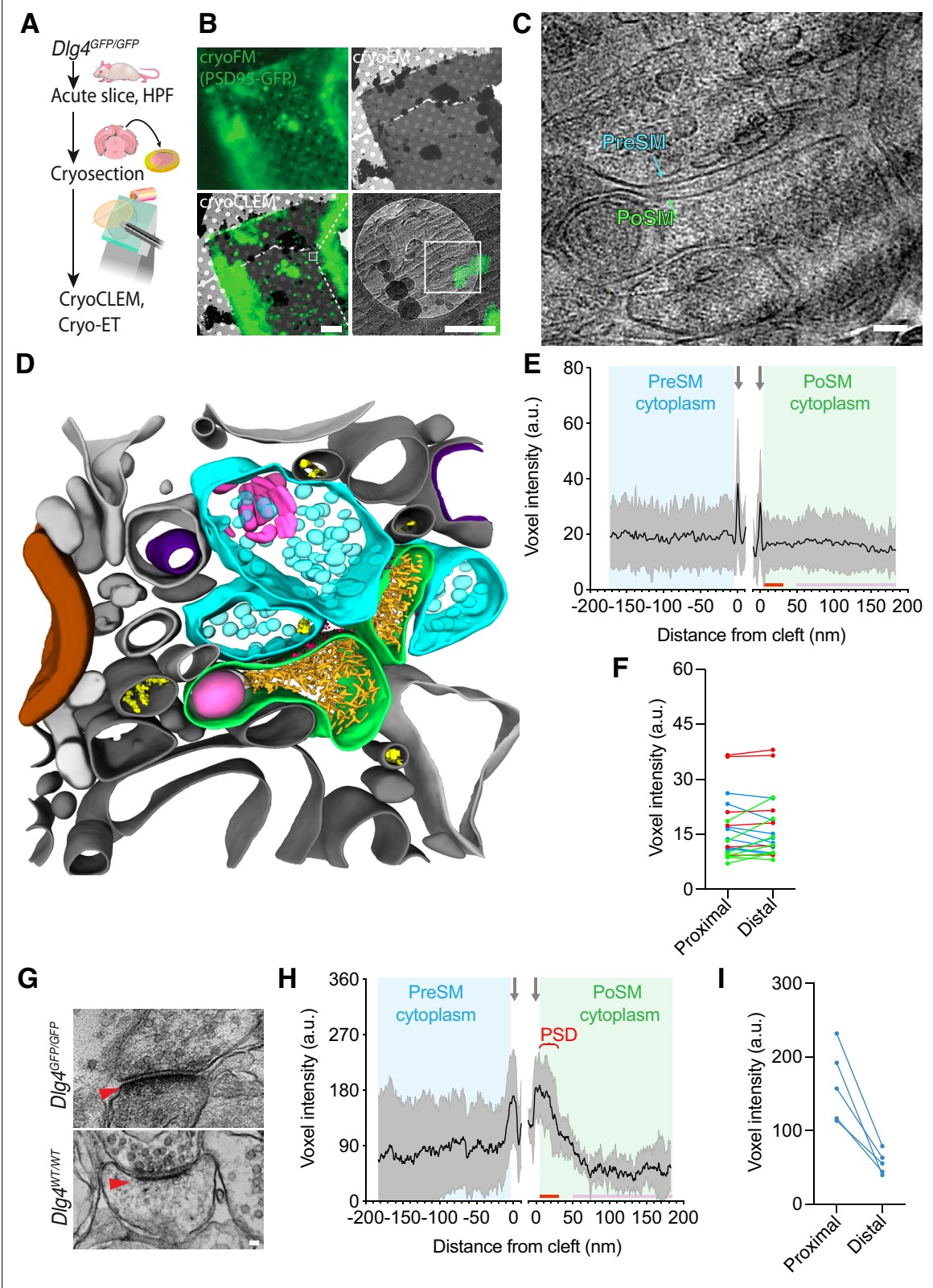

**Figure 2.** The in-tissue molecular architecture of glutamatergic synapses in the adult mammalian brain. (**A**) Schematic showing cryogenic correlated light and electron microscopy (cryoCLEM) and cryo-electron tomography (cryoET) workflow using thin vitreous cryo-sections from forebrains of adult *Dlg4GFP/GFP* knockin mice to determine the in-tissue architecture of glutamatergic synapses. Mice were culled and dissected. 100 μm acute slices were collected, from which 2 mm diameter biopsies of cortex were high-pressure frozen. 70–150 nm thin vitreous cryo-sections were cut from vitrified tissue

*Figure 2 continued on next page*

*Figure 2 continued*

and attached to cryogenic fluorescence microscope (cryoEM) grid for cryoCLEM and cryoET. (**B**) CryoCLEM of a cryoEM grid square containing 150 nm thin vitreous cryo-section. *Top left*, cryoFM image of a holey carbon grid square. *Top right*, cryoEM image of the same grid square shown in top left. *Bottom left*, merged image of cryoFM and cryoEM images indicating location of Psd95-GFP puncta. Scale bar, 5 μm. Black box indicates region enlarged in *Bottom right*, showing Psd95-GFP associated with the PoSM. White box indicates region where images for the tomogram shown in C were acquired. Scale bar, 500 nm. (**C**) Tomographic slice of Psd95-GFP containing synapse within thin vitreous cryo-section of adult mouse cortex. Cyan, PreSM. Green, PoSM. Scale bar, 50 nm. (**D**) 3D segmentation of membranes and macromolecules in a representative tomographic volume of a Psd95-containing glutamatergic synapse within thin vitreous cryo-section of adult mouse cortex. The PoSM (green) was identified by Psd95-GFP cryoCLEM and the PreSM (cyan) was identified by its tethering to the PoSM and the prevalence of synaptic vesicles. Salient organelle and macromolecular constituents are indicated: Magenta, membrane proteins within synaptic cleft. Purple, mitochondrion. Yellow, microtubule. Pink, putative endosomal compartment. Brown, myelin. Grey, vicinal membrane-bound subcellular compartments. (**E**) Molecular profiles same as *Figure 1G* but for 21 in-tissue synapses of vitreous cryo-sections from acute brain slices of seven *Dlg4GFP/GFP* mice. (**F**) Comparison of average molecular density profile in regions 5–30 nm (proximal) and 50–200 nm (distal) from the PoSM of each synapse. Blue, red, and green datapoints, synapses with cytoplasm proximal to the PoSM that contained significantly higher, lower, and not significantly different density from distal regions, respectively (two-tailed *t* test with Bonferroni correction, p<0.05, n=682, and 5673 voxels). (**G**) Conventional EM of synapses in chemically fixed, resin-embedded, heavy metal-stained acute brain slice biopsies from *Dlg4GFP/GFP* knockin (left) and wild-type (right), respectively. Red arrowhead, postsynaptic density. Scale bar, 50 nm. (**H and I**) Same as E and F but for acute brain slice biopsies imaged by conventional EM (chemically fixed, resin-embedded, and heavy metal-stained) from 3 *Dlg4GFP/GFP* mice. PSD, postsynaptic density evident in non-native, conventional EM samples.

The online version of this article includes the following video, source data, and figure supplement(s) for figure 2:

**Source data 1.** spreadsheet detailing the organelle and macromolecular constituents within each tomogram of the in-tissue synapse cryoET dataset, including within the PreSM and PoSM of Psd95-GFP containing synapses and additional subcellular compartments in the vicinity.

**Figure supplement 1.** Molecular density profiling and tomographic slices of 21 in-tissue vitreous cryo-section tomograms containing Psd95-GFP synapses.

**Figure supplement 2.** Molecular density profiling and tomographic slices of in-tissue vitreous cryo-section tomograms containing Psd95-GFP synapses.

**Figure supplement 3.** Molecular density profiling and tomographic slices of in-tissue vitreous cryo-section tomograms containing Psd95-GFP synapses.

**Figure supplement 4.** Molecular density profiling and tomographic slices of in-tissue vitreous cryo-section tomograms containing Psd95-GFP synapses.

**Figure supplement 5.** Molecular density profiling and tomographic slices of in-tissue vitreous cryo-section tomograms containing Psd95-GFP synapses.

**Figure supplement 6.** Molecular density profiling and tomographic slices of in-tissue vitreous cryo-section tomograms containing Psd95-GFP synapses.

**Figure supplement 7.** Conventional EM of adult brain glutamatergic synapses.

**Figure 2—video 1.** Reconstructed tomographic volumes of in-tissue Psd95-GFP-containing synapses.
https://elifesciences.org/articles/100335/figures#fig2video1

**Figure 2—video 2.** Reconstructed tomographic volumes of in-tissue Psd95-GFP-containing synapses.
https://elifesciences.org/articles/100335/figures#fig2video2

in cytoplasmic molecular density that could correspond to a PSD was not a conserved feature of glutamatergic synapses.

To rule out that the absence of a conserved postsynaptic protein dense region was caused by the *Dlg4GFP/GFP* knock-in mutation or the method of sample preparation, we performed conventional EM of acute slice brain samples and ultra-fresh synapses, which showed a PSD in samples from *Dlg4GFP/GFP* comparable to those in wild-type mice (*Figure 2G-H*, *Figure 2—figure supplement 7*). This confirmed that the appearance of the high local protein density that defines the PSD is a consequence of conventional EM sample preparation methods.

## Synaptic organelles

To assess the molecular diversity of adult forebrain excitatory synapse architecture, we next catalogued the identifiable macromolecular and organelle constituents in ultra-fresh synapses (*Figure 1—source data 1*). At least one membrane-bound organelle was present in 61 ± 4% of PoSM compartments and, excluding synaptic vesicles, in 77 ± 2% of PreSM compartments (*Figure 3—figure supplement 1A*). Based on previous characterizations by conventional EM studies (*Cooney et al., 2002*) most membrane-bound organelles were readily separable into three structural classes (*Figure 3—figure supplement 1B*, *Figure 1—source data 1*, see methods): (1) A network of flat, tubular membrane compartments that twisted and projected deep toward the adhered synaptic membrane (*Figure 3A-B*, *Figure 3—figure supplement 1C*). (2) Large spheroidal membrane compartments (>60 nm diameter) that were situated 72–500 nm away from the cleft (*Figure 3A*, *Figure 3—figure supplement 1B, D*).

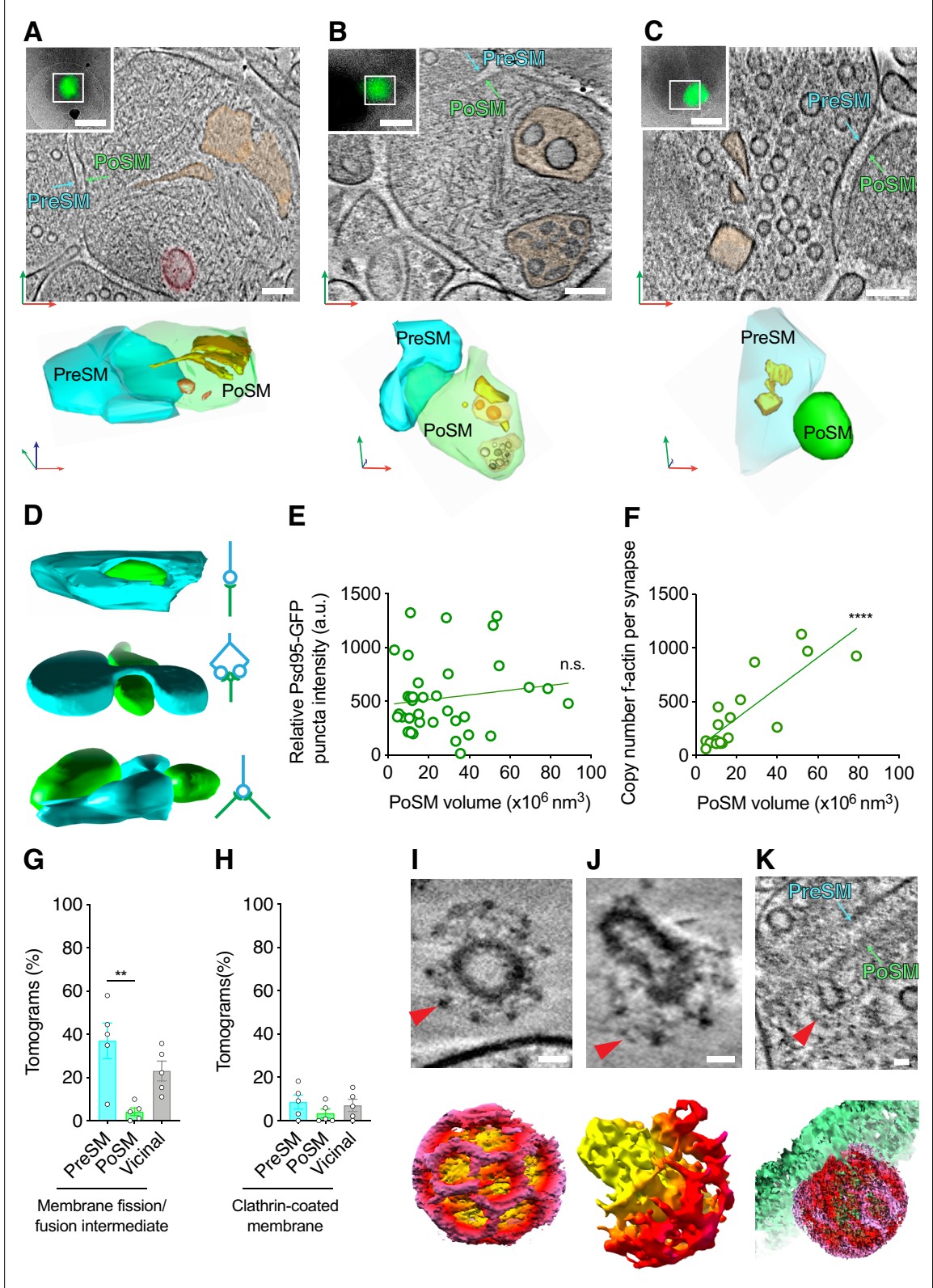

**Figure 3.** Architecture of lipid membrane bilayers in glutamatergic synapses. (**A–C**) Organelles in Psd95-containing synapse shown *top*, as a tomographic slice and *bottom*, 3D segmentation. *Top inset*, Psd95-GFP cryogenic correlated light and electron microscopy (cryoCLEM) image of synapse. Scale bar, 500 nm. Cyan, PreSM. Green, PoSM. Red, green, and blue arrows indicate *x*-, *y*-, and *z*-axis of tomogram, respectively. Scale bar, 100 nm. (**A**) Flat/tubular membrane compartment and large spheroidal membrane compartment pseudo-coloured orange and red, respectively.

*Figure 3 continued on next page*

*Figure 3 continued*

(**B**) Multivesicular bodies pseudo-coloured orange. (**C**) Polyhedral membrane vesicle pseudo-coloured orange (**D**) 3D segmentation (*left*) and schematic (*right*) of Psd95-GFP-containing synapses showing various topologies of connectivity. *Top*, unimodal single input and single output. *Middle*, bifurcated synapse with single input and output. *Bottom*, divergent synapse with single input and two outputs. Cyan, PreSM input. Green, PoSM output. (**E**) Plot of fluorescence intensity (from cryoFM of Psd95-GFP) versus PoSM volume suggesting that there is no correlation between apparent amount of Psd95-GFP and PoSM volume. Linear regression (green line) with Pearson's $r$=0.14, p=0.41. (**F**) Plot showing the copy number of F-actin cytoskeletal elements in PoSM compartment versus PoSM compartment of each synapse. The copy number of actin filaments was estimated from 3D segmented models of 7 nm filaments within the tomographic map. F-actin branching from another filament were counted as a separate filament. Linear regression (green line) with Pearson's $r$=0.83, p<0.0001. (**G–K**) Snapshots and quantification of membrane remodeling within glutamatergic synapses. (**G**) Prevalence of membrane fission/fusion intermediates within PreSM, PoSM, and non-synaptic membranes (vicinal) from tomograms of 5 adult *Dlg4$^{GFP/GFP}$* mice. Error bars, SEM.*, Paired two-tailed *t*-test, p=0.021, n=5 mice. (**H**) Same as G but for clathrin-coated membrane. (**I**) *Top*, tomographic slice of a clathrin-coated (red arrowhead) synaptic vesicle and *bottom*, a 3D tomographic density map for the region shown on top. Clathrin cage and membrane are shown with red and yellow voxels, respectively. Scale bar, 20 nm. (**J**) *Top*, tomographic slice of a clathrin-coat (red arrowhead) encapsulating part of an internal membrane within the PreSM compartment of a Psd95-GFP containing synapse and shown *bottom*, a masked 3D tomographic map. Clathrin cage and membrane are shown with red and yellow voxels, respectively. Scale bar, 20 nm. (**K**) *Top*, tomographic slice, and *bottom*, masked 3D tomographic map showing clathrin-coated endocytic pit (red arrowhead) within the cleft of a Psd95-GFP containing PoSM. Clathrin cage and membrane are shown with red and green voxels, respectively. Scale bar, 20 nm.

The online version of this article includes the following figure supplement(s) for figure 3:

**Figure supplement 1.** Structural categories of membrane-bound organelles in Psd95-containing synapses in the adult brain.

(3) Multivesicular bodies that were positioned throughout the PreSM and PoSM (78–360 nm away from the cleft, *Figure 3B*, *Figure 3—figure supplement 1B, E*). A fourth category of membranous compartments found in our data did not match any of the structures previously reported and can be characterized as polyhedral vesicles with flat surfaces and joined by protein-coated membrane edges and vertices (90° to 35° internal angle, *Figure 3C*, *Figure 3—figure supplement 1F*).

Mitochondria were also readily identified, most abundantly in the presynaptic compartment (44 ± 5% of PreSM) and only once in close proximity to the postsynaptic compartment (*Figure 3—figure supplement 1G*). A striking feature of all presynaptic mitochondria was the presence of amorphous dense aggregates within the mitochondrial matrix (*Figure 3—figure supplement 1H*). Similar aggregates have been identified as solid deposits of calcium phosphate (*Wolf et al., 2017*) and are apparent in cultured neurons (*Tao et al., 2018*). To exclude the possibility that mitochondrial aggregates arose by excitation during sample preparation, we collected tomograms of ultra-fresh synapses prepared in ACSF that lacked divalent cations, including calcium (*Figure 3—figure supplement 1I*). These data showed synapses with mitochondrial aggregates in the absence of exogenous calcium, suggesting they do not arise during sample preparation. Thus, mitochondrial granular aggregates likely represent a physiologically normal feature of the mammalian brain, which is in keeping with the role of synaptic mitochondria as a reservoir of cellular calcium ions (*Billups and Forsythe, 2002*). Overall, the variable protein composition and type of intracellular organelles indicate distinct synapse subtypes or states within the mammalian forebrain.

## Synaptic shape and remodeling

Further diversity of synaptic architectures was evident in the size, shape, and mode of connectivity of the PreSM to the PoSM (*Zhu et al., 2018*). To assess these structural variables, Psd95-GFP-containing PoSMs and apposing PreSMs were semi-automatically segmented (see methods). This revealed synapses with multiple different topologies, including with single, double, and divergent (1 PreSM: 2 PoSM) synaptic contacts (*Figure 3D*, *top*, *middle*, and *bottom*, respectively, and *Figure 1—source data 1*). These were consistent with the known diversity of cellular connectivity mediated by glutamatergic synapses (*Harris and Weinberg, 2012*). The volume of the PoSM compartment in our dataset ranged from 3x10⁶ to 79x10⁶ nm³. In keeping with the known diversity of Psd95-containing synapses (*Zhu et al., 2018*) and in vivo fluorescence imaging experiments (*Melander et al., 2021*), the size of PoSM compartments was not correlated with the amount of Psd95-GFP detected by cryoCLEM (p=0.7, Pearson's $r$=0.06, *Figure 3E*; *Melander et al., 2021*). However, the size of PoSM compartments was correlated with the number of actin cytoskeletal filaments (p<0.0001, Pearson's $r$=0.83, *Figure 3F*), which is in line with the apparent non-uniform distribution of F-actin in distinct sub-cellular compartments (*Figure 1E*) and the role of F-actin in regulating postsynaptic geometry and structural plasticity (*Okamoto et al., 2007*).

The variation in size, shape, and molecular composition of synaptic membranes is thought to arise by membrane remodeling, including exo- and endocytosis. In 32 ± 6% synapses we identified pit-like, hemifusion or hemifission structures in the membrane (*Figure 3—figure supplement 1J*) that likely indicate intermediates of membrane remodeling trapped within the sample at the moment of freezing (*Figure 1—source data 1*). Significantly more remodeling intermediates were identified in the PreSM compared to PoSM compartments (*Figure 3G*, Paired two-tailed *t*-test, p=0.021, n=5 mice), which is consistent with the expected high frequency of membrane dynamics associated with synaptic vesicle recycling in this compartment (*Kononenko and Haucke, 2015*).

In a subset of tomograms (21 ± 11%, *Figure 3H*), a clathrin coat was unambiguously identified by the triskelion-forming pentagonal and hexagonal openings (*Kanaseki and Kadota, 1969*; *Kravčenko et al., 2024*) evident in the raw tomographic maps (*Figure 1—source data 1*). In the pre-synaptic compartment, clathrin surrounded synaptic vesicles (*Figure 3I*) and budding vesicles on internal membrane compartments (*Figure 3J*), which is in agreement with previous time-resolved EM experiments (*Watanabe et al., 2014*). In the postsynaptic compartment, a clathrin-coated pit formed directly on the cleft, suggesting membrane and cargo can be removed without prior lateral diffusion out of the synapse (*Figure 3K*). Thus, the distribution of these distinct intermediates suggests that membrane remodeling occurs at particular sub-compartments of the synapse (*Borges-Merjane et al., 2020*; *Imig et al., 2020*; *Watanabe et al., 2014*).

## Synaptic cleft and ionotropic glutamate receptors

Geometric properties of each synapse are thought to be critical for determining synaptic strength, including synaptic cleft height (*Cathala et al., 2005*; *Rusakov et al., 2011*), measured as the shortest distance between PreSM and PoSM. To quantify this variable in our dataset, we computationally determined the 3D coordinates of the cleft within each synapse (*Figure 4—figure supplement 1A* and see methods). The PreSM to PoSM distances across the cleft indicated an average cleft height of 33 nm (*Figure 4A*). These values differ from average cleft heights of ~20 nm determined in non-native conditions by conventional EM (*Peters et al., 1970*) but are consistent with super-resolution light microscopy imaging (*Tønnesen et al., 2018*). Interestingly, the average cleft height of each synapse in our dataset varied (ranging from 27 to 37 nm). A similar range of cleft distances were observed in in-tissue synapse tomograms (*Figure 2C*, *Figure 2—figure supplements 1–6*), suggesting that the cleft dimensions were largely unaffected by the ultra-fresh sample preparation. A subset of 'ultra-fresh' synapses maintained a very broad or bimodal distribution of cleft heights (*Figure 4—figure supplement 1B*). Inspection of synapses with a broad or bimodal distribution revealed closely apposed and remotely spaced subsynaptic regions of the cleft, containing transsynaptic complexes with varying dimensions that complemented the cleft height of the subregion (*Figure 4—figure supplement 1C–D*). Closely opposed synaptic subregions were found both near the edge and the middle of the cleft, consistent with the notion that cleft height is locally regulated. Thus, these data suggest the synaptic cleft as a highly plastic interface, in which cleft dimensions are dictated by the variable molecular composition of transsynaptic adhesion complexes (*Missler et al., 2012*).

The number, concentration, and distribution of ionotropic glutamate receptors anchored at the postsynaptic membrane by Psd95 is a central determinant of synaptic output (*Béïque et al., 2006*; *Cathala et al., 2005*; *Kerr and Blanpied, 2012*; *Migaud et al., 1998*; *Rusakov et al., 2011*; *Tang et al., 2016*). Putative ionotropic glutamate receptors were readily identified in 95 ± 2% of PoSM compartments. These resembled the side (*Figure 4B left*) and top views (*Figure 4B right*) of ionotropic glutamate receptor atomic models (*Hansen et al., 2021*; *Regan et al., 2015*; *Zhu and Gouaux, 2017*). To confirm the identity of these proteins, we manually picked 2,522 receptors and extracted sub-volumes at those positions. We then applied subtomogram averaging procedures (see methods), which gave a 25 Å resolution 'Y' shaped cryoEM density map extending 14 nm from the PoSM (*Figure 4—figure supplement 2A*). The density shows a twofold symmetric structure containing two layers of globular domains in positions proximal and distal to the PoSM (*Figure 4C*) as expected for the structure of an ionotropic glutamate receptor. Consequently, the atomic structure of the AMPA receptor was well accommodated by the density, which supports the identification of these ion channels within our synapse tomogram dataset (*Figure 4C*).

Next, to analyse the 3D configuration of populations of ionotropic glutamate receptors, we used these coordinates to quantify receptor clusters (see methods). Clusters were identified in 78% of

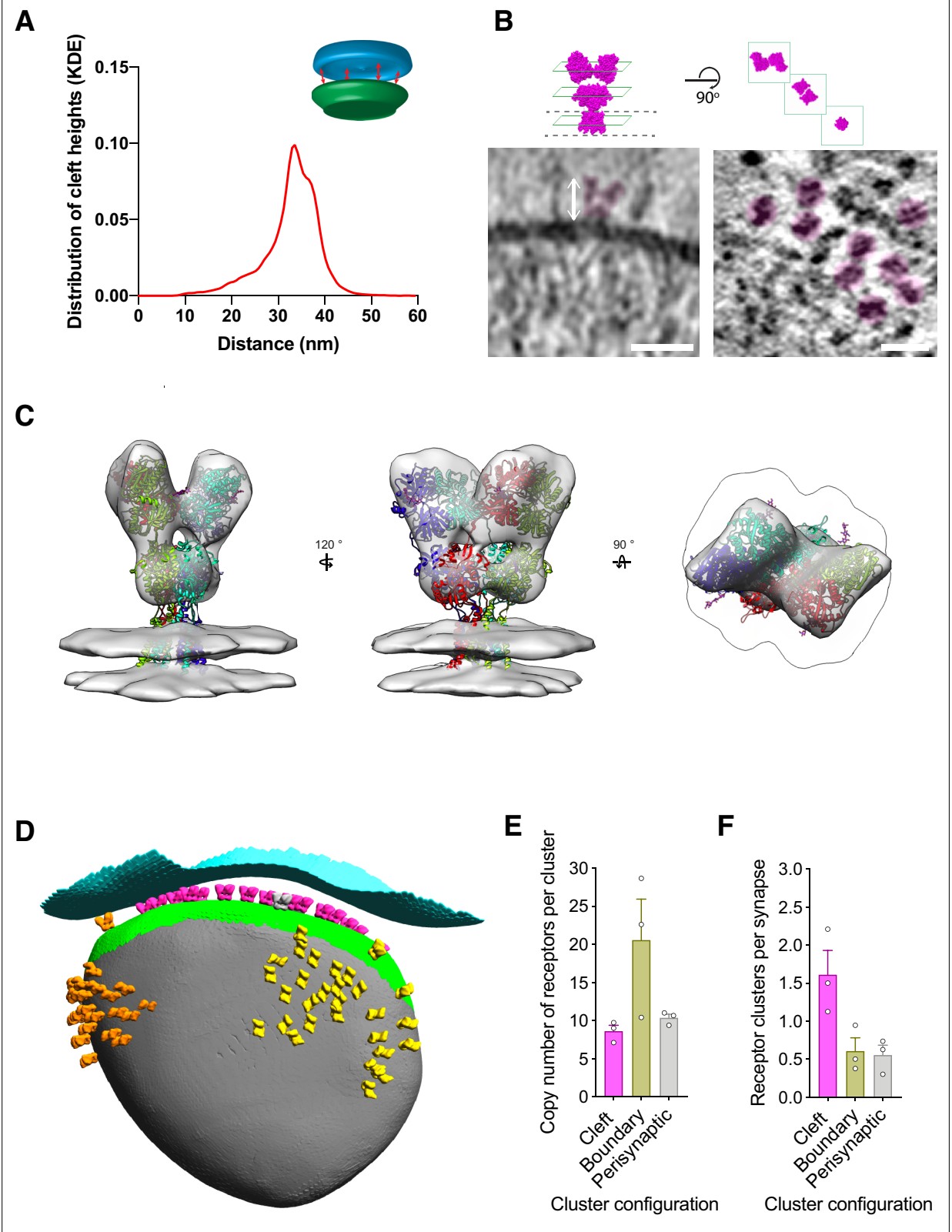

**Figure 4.** Structural variables of synaptic strength. (**A**) Distribution of cleft height distances of all synapses is shown as a kernel density estimation (KDE). *Inset*, schematic depicting nearest neighbour method for defining the synaptic cleft between pre- (PreSM, cyan) and post-synaptic membrane (green) compartments. (**B**) *Top left*, 'side' view of atomic structure of AMPA subtype of ionotropic glutamate receptor (PDB ID: 3kg2). Dashed line, boundary of transmembrane domain. *Top right*, 'top' views of distal amino-terminal domain layer, proximal ligand-binding domain layer, and transmembrane domain

*Figure 4 continued on next page*

*Figure 4 continued*

layer. *Bottom left*, tomographic slice of Psd95-GFP containing membrane oriented approximately parallel to the electron beam. Putative ionotropic glutamate receptor pseudo-coloured in magenta. *Bottom right*, tomographic of Psd95-GFP containing membrane oriented approximately orthogonal to the electron beam. Multiple putative ionotropic glutamate receptor pseudo-coloured in magenta. Scale bar, 20 nm. (**C**) In situ structure of ionotropic glutamate receptors within Psd95-GFP containing PoSM compartments determined by subtomogram averaging of 2368 sub-volumes. Atomic model of an AMPA receptor (GluA2, PDB: 3kg2) docked in the determined map. Subunits of the model are coloured in red, yellow, cyan, and purple. (**D**) Cluster analysis of identified ionotropic glutamate receptors depicted as a 3D model of the postsynaptic compartment with ionotropic glutamate receptors in the orientation and position determined by subtomogram averaging. Each cluster is coloured differently: magenta, orange, and yellow. Presynaptic and postsynaptic cleft membrane are shown in cyan and green, respectively. The postsynaptic membrane outside of the cleft is shown in grey.

(**E**) Receptor number per cluster for three different cluster locations: in the cleft (magenta), partly inside and outside the cleft (boundary, yellow), and completely outside the cleft (perisynaptic, grey) from tomograms of three adult *Dlg4*$^{GFP/GFP}$ mice. (**F**) Prevalence of receptor clusters for three different cluster locations: in the cleft (magenta), partly inside and outside the cleft (boundary, yellow), and completely outside the cleft (perisynaptic, grey) from tomograms of three adult *Dlg4*$^{GFP/GFP}$ mice.

The online version of this article includes the following figure supplement(s) for figure 4:

**Figure supplement 1.** Quantification and variability of cleft height in glutamatergic synapses.

**Figure supplement 2.** Subtomogram averaging and anchoring of ionotropic glutamate receptors.

synapses with an average of 2 clusters per synapse (*Figure 4D*), which were composed of up to 60 ionotropic glutamate receptors (average 10 receptors per cluster) (*Figure 4E*). These subsynaptic regions with higher concentrations of ionotropic glutamate receptors were reminiscent of receptor 'nanodomains' detected by super-resolution microscopy (*Broadhead et al., 2016*; *Dani et al., 2010*; *MacGillavry et al., 2013*; *Nair et al., 2013*; *Tang et al., 2016*).

The position of ionotropic receptor clusters in synaptic and extra-synaptic sites, particularly the most abundant AMPA receptor subtype, is expected to impact synaptic strength and play an important role in synaptic plasticity (*Park, 2018*). We, therefore, analyzed the distribution of clusters in the context of the overall synapse architecture, which revealed that 60% of synapses maintained multiple large clusters of ionotropic glutamate receptors completely or partly outside of the cleft (*Figure 4D–F*). The apparent detection of perisynaptic populations of glutamate receptors resemble the distributions observed by previous freeze-fracture immunogold EM (*Masugi-Tokita and Shigemoto, 2007*) and super-resolution light microscopy (*Nair et al., 2013*). The close proximity of perisynaptic clusters to the cleft is consistent with the functional necessity of a non-cleft population of ionotropic glutamate receptors that are recruited into the cleft during the early phase of long-term potentiation (*Penn et al., 2017*). Interestingly, the cytoplasmic C-termini of ionotropic receptor clusters both inside and outside of the cleft were situated adjacent to an elaborate and heterogeneous membrane- and actin-associated machinery (*Figure 4—figure supplement 2B*), possibly indicating that a structural mechanism contributes to the positioning of receptors both inside and outside the synaptic cleft.

## Discussion

Here, we determined the near-physiological 3D molecular architecture of glutamatergic synapses in the adult mammalian brain. In both ultra fresh and anatomically intact in-tissue cryoET datasets, a conserved relatively higher molecular density consistent with a PSD was absent. Instead, we observed subtle variations in molecular crowding within our cryoET dataset. The PSD has been a defining structure of glutamatergic synapses and has been most frequently observed by conventional EM (*Bourne and Harris, 2008*; *Sheng and Kim, 2011*), including when using more recent methods that involve freeze-substitution into organic solvents, followed by heavy-metal staining (*Chen et al., 2008*). We addressed the hypothesis that the occurrence of a PSD might be related to sample preparation and staining techniques of conventional EM by preparing the same mouse brain samples for conventional EM and were indeed able to reproduce the conserved occurrence of a PSD. We, therefore, suggest that the appearance of a PSD in room temperature EM experiments could arise during sample preparation. Preferential heavy-metal staining, crosslinking of interacting proteins by fixatives, or extraction of cytoplasmic proteins that are not directly or indirectly associated with the postsynaptic membrane could alter the apparent local distribution or relative density of proteins in tissue samples prepared by resin-embedding and freeze-substitution EM methods.

CryoEM of in vitro cultured organotypic slice cultures that first demonstrated the use of high-pressure freezing neuronal tissue samples, revealed synapses with various pre- and post- and trans-synaptic regions containing relatively higher densities (*Zuber et al., 2005*), albeit these data also lacked a label to confirm which synapses in the sample were glutamatergic.

Pioneering cryoET work of synapses purified from brain (synaptosomes) suggested a slightly higher relative molecular density juxtaposed to the postsynaptic membrane (*Fernández-Busnadiego et al., 2010*). However, interpretation of synaptosome architectures may be compromised by lengthy sample preparation in non-physiological solutions that may cause a loss of cytoplasm and the native molecular architecture. Indeed, the marked absence of microtubules in synaptosomes is indicative of sample deterioration. It was also previously challenging to identify glutamatergic synapses definitively. Consequently, synaptosomes that lacked an apparent postsynaptic thickening were actively excluded from data collection (*Martinez-Sanchez et al., 2021*).

More recent cryoCLEM and cryoET studies of primary neuronal cell cultures identified glutamatergic synapses with some but not all showing a subtly higher concentration of macromolecules juxtaposed to the PoSM relative to distal regions deep within the postsynaptic compartment (*Tao et al., 2018*). These data were consistent with ultra-fresh synapse tomograms from adult brain, including the preservation of microtubules in the presynaptic compartment. A greater fraction of ultra-fresh adult synapses resembled the minority of primary neuronal synapses that lacked a relative increase in molecular crowding at the PoSM. In anatomically intact brain slice preparations that were nearer to the physiological state of adult brain, even fewer instances of synapses were observed with a relative increase in molecular crowding of the cytoplasm at the PoSM. We suggest that these subtle differences of molecular crowding are likely a consequence of synapse heterogeneity (*Zhu et al., 2018*) or the source, in vitro cell culture versus adult brain, of central synapses. More broadly, these cryoET data were consistent in suggesting that a relatively higher concentration of proteins juxtaposed to the PoSM is neither conserved nor a defining feature of glutamatergic synapses.

The synaptic architectures reported here were nevertheless consistent with the existence of specialized protein nanodomains of particular synaptic constituents detected by fluorescence labeling, including Psd95 (*Broadhead et al., 2016*; *Dani et al., 2010*; *MacGillavry et al., 2013*; *Nair et al., 2013*; *Tang et al., 2016*). Indeed, clusters of ionotropic glutamate receptors that are anchored via interactions with Psd95 (*Dani et al., 2010*; *MacGillavry et al., 2013*; *Nair et al., 2013*; *Tang et al., 2016*) were a salient feature in tomographic maps of adult brain synapses.

We also evaluated the dimensions of the synaptic cleft, which is expected to affect synaptic strength (*Rusakov et al., 2011*). Cleft height was highly variable and on average 65% greater than that observed by conventional EM, which is also likely explained by the harsh treatments and the well-characterised shrinkage associated with chemical fixation of tissues in conventional EM. More broadly, these comparisons highlight the advantages of combining recent advances in cryoET (*Turk and Baumeister, 2020*) with genetic knock-in labelling to determine the molecular architecture of fresh tissues.

Notwithstanding the absence of a conserved PSD, the arrangement of prominent constituents, including branched F-actin networks, organelles, ionotropic glutamate receptors, and transsynaptic adhesion complexes, appeared to define the architecture of each glutamatergic synapse. Overall, the apparent variability of molecular and membrane architectures, from one synapse tomogram to the next, is in keeping with the enormous diversity of glutamatergic synapse types in the mammalian brain (*Zhu et al., 2018*). Since submission of our manuscript, several reports of synapse cryoET from within cultured primary neurons (*Held et al., 2024a*; *Held et al., 2024b*) and mouse brain (*Glynn et al., 2024*; *Matsui et al., 2024*) were prepared by cryoFIB-milling. These new datasets are largely consistent with the data reported here. While cryoFIB-milling has the advantage of overcoming the local knife damage caused by cryo-sectioning, it introduces amorphization across the whole sample that diminishes the information content (*Al-Amoudi et al., 2005*; *Lovatt et al., 2022*; *Lucas and Grigorieff, 2023*). We and others have recently shown that cryoET data from vitreous cryo-sections can reveal in-tissue protein structures at subnanometer and near-atomic resolution (*Gilbert et al., 2024*; *Elferich et al., 2025*). A future challenge will be to decipher how synaptic structural variability across the dendritic tree, different brain regions, and neuronal subtypes determine specific functions within the mammalian brain.

## Methods

### Mouse genetics and breeding

Animals were treated in accordance with the UK Animal Scientific Procedures Act (1986) and NIH and ARRIVE guidelines. All animal experiments were approved by the University of Leeds Animal Welfare and Ethics Committee. The generation and characterization of the Psd95-GFP knockin mouse (*Dlg4*<sup>GFP</sup>) was described (*Broadhead et al., 2016*; *Zhu et al., 2018*). The endogenous Psd95 protein levels, assembly into supercomplexes, anatomical localization, developmental timing of expression and electrophysiological function in the hippocampus CA1-CA3 synapses in *Dlg4*<sup>GFP</sup> mice are indistinguishable from WT mice (*Broadhead et al., 2016*; *Frank et al., 2016*; *Zhu et al., 2018*).

### Ultra-fresh synapse preparation

Samples were prepared from 5 adult (P65-P100) male *Dlg4*<sup>GFP/GFP</sup> knockin mice. Sample preparation from culling to cryopreservation of each mouse was less than 2 min. Mice were culled by cervical dislocation. Forebrain, including cortex and hippocampus, that encompass the known diversity gluta-matergic synapses (*Zhu et al., 2018*) were removed (in <40 s) and were homogenized (<20 s) by applying 12 strokes of a Teflon-glass pestle and mortar containing 5 ml ice-cold (~1 °C) carbogenated (5% $CO_2$, 95% $O_2$) artificial cerebrospinal fluid (ACSF; 125 mM NaCl, 25 mM KCl, 25 mM NHCO$_3$, 25 mM glucose, 2.5 mM KCl, 2 mM CaCl$_2$, 1.25 mM NaH$_2$PO$_4$, 1 mM MgCl$_2$; Osmolality: 310 mOsM/L). Ice-cold ACSF was used to limit mechanical stimulation since synaptic transmission and other enzyme-catalyzed processes (e.g. endocytosis) are negligible at this temperature (*Volgushev et al., 2000*). 1 µl homogenate was diluted into 100 µl ice-cold ACSF containing 1:6 BSA-coated 10 nm colloidal gold (BBA; used as a fiducial marker) (<10 s). 3 µl sample was applied onto glow-discharged 1.2/1.3 carbon foil supported by 300-mesh Au or Cu grids (Quantifoil) held at 4 °C, 100% humidity in Vitrobots (Mark IV, Thermo Fisher). Excess sample was manually blotted from the non-foil side of the grid for ~4 s with Whatman paper (No. 1; manipulated within the Vitrobot with tweezers) before cryopreserving by plunge freezing in liquid ethane held at –180 °C (*57*). 1–2 grids were prepared from each mouse using 1–2 Vitrobots.

### In-tissue acute slice synapse preparation

Samples were prepared from 6 adult (P65-P100) male *Dlg*<sup>GFP/GFP</sup> knockin mice. Mice were culled by cervical dislocation. Forebrain, including cortex and hippocampus, were removed and 100 µm coronal acute slices were prepared in cutting buffer (93 mM N-methyl-D-glucamine, 2.5 mM KCl, 1.2 mM NaH2PO4, 30 mM NaHCO3, 20 mM HEPES, 5 mM Sodium ascorbate, 2 mM Thiourea, 3 mM Sodium Pyruvate, 10 mM MgSO4, 0.5 mM CaCl2, pH 7.39–7.40, osmolarity 305–315 mOsm/L) using a Leica VT1200S vibratome at 0.5–2°C. Tissue biopsies of acute slices were collected following the method established by Zuber and co-workers in which samples remain viable (*Zuber et al., 2005*). Briefly, slices were recovered in ACSF perfused with carbogen at room temperature for 45 minutes. Next, biopsies were collected with a 2 mm biopsy punch and were transferred to carbogenated NMDG cutting buffer supplemented with 20% dextran 40,000. Biopsies were high-pressure frozen within cryoprotectant into 3 mm carriers using a Leica EM ICE. Tissue was cut into 70–150 nm thick vitreous cryo-sections using a Leica FC7 cryo-ultramicrotome with a CEMOVIS diamond knife (Diatome) from regions of the tissue that were ~25 µm from the vibratome cutting edge. Vitreous cryo-sections were attached to cryoEM grids at –150 °C.

### Cryogenic correlated light and electron microscopy

Cryogenic fluorescence microscopy (cryoFM) was performed on the Leica EM cryoCLEM system with a HCX PL APO 50 x cryo-objective with NA = 0.9 (Leica Microsystems), an Orca Flash 4.0 V2 sCMOS camera (Hamamatsu Photonics), a Sola Light Engine (Lumencor) and the following filters (Leica Microsystems): green (L5; excitation 480/40, dichroic 505, emission 527/30), red (N21; excitation 515–560, dichroic 580, emission LP 590), and far-red (Y5; excitation 620/60, dichroic 660, emission 700/75). During imaging, the humidity of the room was controlled to 20–25% and the microscope stage was cooled to –195 °C. Ice thickness was assessed with a brightfield image and L5 filter. Regions of the grid with thin ice (~50%) were imaged sequentially with the following channel settings: 0.4 s exposure, 30% intensity in green channel; 40 ms exposure, intensity 11% in BF channel; 0.4 s exposure, 30% intensity in red channel, 0.4 s exposure, 30% intensity in far-red channel. 0.3 µm separated

z-stack of images was collected for each channel over a 5–20 µm focal range. Images were processed in Fiji. The location of Psd95-GFP was evident as puncta of varying brightness that were exclusively in the green channel. The occasional autofluorescent spot was identifiable by fluorescence across multiple channels, including green and red channels. Grid squares were selected for cryoCLEM that contained Psd95-GFP puncta within the holes of the holey carbon grid.

## Cryogenic electron microscopy and tomography

CryoEM and CryoET data collection was performed on a Titan Krios microscope (FEI) fitted with a Quantum energy filter (slit width 20 eV) and a K2 direct electron detector (Gatan) running in counting mode with a dose rate of ~4 e⁻/pixel/second at the detector level during tilt series acquisition. Intermediate magnification EM maps of the selected grid squares were acquired at a pixel size of 5.1 nm. Using these intermediate magnification maps of each grid square and the corresponding cryoFM image, the location of Psd95-GFP puncta was manually estimated; performing computational alignment of cryoFM and cryoEM images before tomogram acquisition was not necessary.

An unbiased tomographic data collection strategy was followed, guided by the location Psd95-GFP cryoCLEM data. 93 ultra-fresh and 50 acute slice/in-tissue (vitreous cryo-section) Psd95-GFP synapse tomograms were collected from 5 and 6 mice, respectively. No Psd95-GFP tomograms were excluded from the dataset (*Figure 1—source data 1*, *Figure 2—source data 1*). Tomographic tilt series of the cryoFM-correlated Psd95-GFP locations were collected between ±60° using a grouped dose-symmetric tilt scheme (*Hagen et al., 2017*) in SerialEM (*Mastronarde, 2005*) and a phase plate (*Fukuda et al., 2015*) pre-conditioned for each tomogram. Groups of 3 images with 2° increments were collected. Images with a 2 s exposure with 0.8–1.25 µm nominal defocus at a dose rate of ~0.5 e⁻/Å/s were collected as 8x0.25 s fractions, giving a total dose of ~60 e⁻/Å over the entire tilt series at a calibrated pixel size of 2.89 Å. This pixel size corresponds to a field of view of 1.3 µm². Consequently, the centres of all synapses were captured in the tomographic dataset and synapses with a maximum dimension under 1.3 µm were contained entirely within the tomogram. Tomograms of thin vitreous cryo-sections were collected with the same protocol, but replacing the phase plate with a 100 µm objective aperture and acquiring tilt series images at 5–6 µm nominal defocus with a nominal pixel size of 3.42 Å.

## Conventional EM

Samples were prepared from adult (P65-P100) male *Dlg4^GFP/GFP* knockin and wild-type mice from three sources: (i) Fresh 2 mm diameter biopsy samples of the primary or secondary somatosensory cortex were collected from acute brain slices (as described above) and were fixed with chemi-fix buffer composed of 4% paraformaldehyde 2% glutaraldehyde in 50 mM phosphate (PB) pH7.4 for 1 hr. (ii) Ultra-fresh synapses were prepared (as described above) and incubated in chemi-fix buffer for 40 min at room temperature. (iii) Anesthetized mice were cardiac perfused with 20 ml PB, followed by 40 ml chemi-fix buffer. 100 µm coronal sections were collected from the whole brain with a vibratome, from which 2 mm biopsy samples of the cortex were collected and incubated in chemi-fix buffer at 4°C overnight.

All samples were next washed three times in PB for 1 min before post-fixation in 2% osmium tetroxide for 1 hr at room temperature. Next, samples were washed three times in PB for 1 min each and once in 50% ethanol for 1 min. Samples were next stained with 4% w/v uranyl acetate in 70% for 1 hr at room temperature before dehydration with sequential washes in 70% ethanol, 90% ethanol, 100% ethanol, and 100% acetone for 5 min each. Samples were resin-embedded by washing twice with propylene oxide for 20 min before incubation in 50% araldite epoxy resin in propylene oxide at room temperature overnight. Finally, samples were incubated twice in epoxy resin for 10–16 hr before the resin was polymerized in a mold at 60 °C for 3 days. Ultrathin (80–100 nm) sections collected from an ultramicrotome were placed on 3.05 mm copper grids, stained with saturated uranyl acetate for 30 min, and Reynold's lead citrate for 5 min, and imaged on a Tecnai F20 TEM at 5000 x and 29,000 x magnification.

## CryoCLEM image processing

Computational correlation between the cryoFM and cryoEM images was conducted using custom Matlab (Mathworks) scripts as described previously (*Kukulski et al., 2011*; *Schorb and Briggs,*

*2014*). The centre points of at least 10 holes in the holey-carbon foil around each Psd95 position were used as fiducial markers to align the green channel cryoFM image to a montaged intermediate magnification EM image covering the whole grid square. All GFP puncta correlated within membrane-bound compartments with an apparent presynaptic membrane attached to the postsynaptic membrane. The identity of presynaptic membranes was confirmed by the presence of numerous synaptic vesicles.

To quantify relative amount of Psd95 in each synapse, Psd95-GFP puncta in cryoFM images were segmented using the watershed algorithm in ImageJ, from which the pixel intensities of each puncta were integrated.

## Tomogram reconstruction

Frames were aligned and ultra-fresh synapse tomograms were reconstructed using 10–20 tracked 10 nm fiducial gold markers in IMOD (*Kremer et al., 1996*). Tomograms of synapses in thin vitreous cryo-sections were aligned by patch tracking and denoised using SPIRE-crYOLO (*Wagner et al., 2019*). Tomograms used for figures, particle picking, annotation of macromolecular constituents, and molecular density analysis were generated with five iterations of SIRT and binned to a voxel size of 11.94 Å (binned 4 x). Weighted back-projection tomograms (binned 4 x and 2 x) were used for subtomogram averaging (see below).

## Annotation and analysis of macromolecular constituents in tomograms

Annotation described in *Figure 1—source data 1* was performed blind by two curators to assess ultra-fresh synapse structures in IMOD. First, each curator annotated all SIRT reconstructed tomograms independently. Next, a third curator inspected and certified each annotation. The PoSM was identified using Psd95-GFP cryoCLEM (see above). The PreSM was identified as the membrane compartment attached to the PoSM via transsynaptic adhesion proteins and containing numerous synaptic vesicles. Docked synaptic vesicles were defined as vesicles connected 2–8 nm from the PreSM via macromolecular tethers. The average diameter of synaptic vesicles was 40.2 nm and the minimum and maximum dimensions ranged from 20–57.8 nm, measured from the outside of the vesicle that included ellipsoidal synaptic vesicles similar to those previously reported (*Tao et al., 2018*). To assess potential mechanical damage to the native architecture of ultra-fresh synapses, the PoSM or PreSM compartments were categorized as either open or closed. An open PreSM or PoSM compartment indicated that the plasma membrane was ruptured, whereas closed PreSM and PoSM compartments were similar to structures obtained from vitreous cryo-sections, indicating the native architecture was retained. Only synapse tomograms with closed PreSM or PoSM were used for further analysis of the molecular architecture and constituents of synapses.

Organelles, intermediates of membrane fission or fusion, and identifiable macromolecules were annotated within the presynaptic, postsynaptic and non-synaptic (vicinal) compartments within the tomogram dataset (*Figure 1—source data 1*), which were classified as follows: (1) Flat/tubular membrane compartments: Flat membrane tubules 12–20 nm wide with ≥180° twist along the tubule axis. (2) Large spheroidal membrane compartments: Vesicles with a diameter greater than 60 nm. (3) Multivesicular bodies: Membranous compartments containing at least one internal vesicle. (4) Polyhedral vesicles: Membranous compartment composed of flat surfaces related by ≤90° angle. (6) Dense-cored vesicles. (7) Fission/fusion intermediates: hemifusion intermediates, protein-coated invaginations of membrane. (8) Clathrin-coated intermediates: invaginations with a protein coat containing pentagonal or hexagonal openings. (9) Mitochondria: double membrane with cristae and electron-dense matrix. (10) Mitochondrial intermediates: mitochondria with budding outer membrane or putative mitochondrial fragments wrapped by two membrane bilayers. (11) Ribosomes: 25–35 nm electron-dense particles composed of small and large subunits. (12) F-actin: ~7 nm diameter helical filaments. In the PoSM, F-actin formed a network with ~70° branch points (*Figure 1—figure supplement 1C*) likely formed by Arp2/3, as expected (*Fäßler et al., 2020*; *Pizarro-Cerdá et al., 2017*). Putative filament copy number in the PoSM was estimated by manual segmentation in IMOD. (13) Microtubules: ~25 nm diameter filaments. (14) Cargo-loaded microtubules: vesicles attached to microtubules via 25–30 nm protein tethers.

## Molecular density profile analysis

Molecular density analysis *Smith and Langmore, 1992* of the PreSM and PoSM compartments from ultra-fresh synapse and in-tissue vitreous cryo-section tomograms was carried out using voxel intensity line profiles of the presynaptic and postsynaptic membrane measured using IMOD and Fiji. Only Psd95-GFP-labelled synapse tomograms were analysed and tomograms were only excluded from analysis if they were oriented on the missing wedge, or if ice contamination, EM grid carbon or cutting damage obscured the PoSM cytoplasm (see *Figure 1—figure supplements 3–9*, *Figure 2—figure supplements 1–6* showing tomographic slices of each ultra-fresh and in-tissue synapse tomogram, respectively). All profiles were made at central regions of each synapse, containing docked presynaptic vesicles and clusters of ionotropic glutamate receptors (*Figure 1—figure supplements 3–9*), because this is where the PSD is consistently located in synapses imaged by conventional EM. Line profiles (36 nm wide and 450 nm long) from 31 single tomographic slices from synapses with PoSM and PreSM oriented on the *z*-axis of the tomographic volume were measured. Note, projecting multiple tomographic slices, often used to analyse molecular density of the synapse, will only provide accurate profiles of the cytoplasm if the PoSM is perfectly oriented in the *z*-direction of the tomographic volume. Inspecting the line profiles of each tomographic slice showed that the majority tomograms contained synapses that were slightly tilted with respect to the *z*-axis (*Figure 1—figure supplement 2A–B*). We observed that projections of these synapses caused the PoSM membrane to smear resulting in erroneous cytoplasmic profiles that contained molecular density from the membrane itself (*Figure 1—figure supplement 2C*). This highlighted a risk of projecting multiple tomographic slices to measure molecular density within a specific compartment. We, therefore, avoided directly generating a 2D projection of multiple tomographic slices of each tomographic volume. Instead, to avoid this error, individual profiles of each tomographic slice were aligned to the plasma membrane peak before profiles were averaged to estimate the relative density of molecular crowding within subvolumes of the PreSM and PoSM compartments of each synapse (*Figure 1—figure supplement 2D*). Tomographic slices in which the PoSM was tilted in the *x-y* plane relative to the majority of slices were also excluded from the profile (*Figure 1—figure supplement 2B*, **top panel**). To ensure molecular density analysis was capable of detecting a PSD, similar line profiles were taken from samples prepared for conventional EM (see sample preparation details below, *Figure 2H*, *Figure 2—figure supplement 7*) as a positive control.

## Quantitative analysis of the PreSM, PoSM, and synaptic cleft

Each membrane was segmented in Dynamo (*Castaño-Díez et al., 2017*), generating 4000–20,000 coordinates for each membrane. To refine the contouring of the membrane, subtomograms of the membrane were refined against an average. The synaptic cleft was defined computationally using a custom Matlab-based script that identifies the nearest neighbour coordinates between PreSM and PoSM, and vice versa. This gave a dataset of cleft height distances for each synapse, in which more than 99% of measurements were distributed within a range of 10–45 nm and none were greater than 60 nm. The distribution of cleft height distances was plotted using Kernel Density Estimation in MATLAB.

## Subtomogram averaging

Subtomogram alignment and averaging was performed using Matlab-based scripts derived from the TOM (*Nickell et al., 2005*) and Av3 (*Förster et al., 2005*) toolboxes, essentially as previously described (*Wan et al., 2017*) using initial datasets from only the first 3 mice.

Coordinates of putative ionotropic glutamate receptors were manually picked (2,522 subtomograms) in UCSF Chimera (*Pettersen et al., 2004*) using two-times binned tomograms with an effective voxel size of 11.94 Å. All membrane proteins with a long axis extending ~14 nm out of the membrane and an apparent two-fold symmetry axis were picked. The orientation of the normal vector to the membrane surface was determined to define the initial orientation of each subtomogram. The Euler angle defining the final in-plane rotation angle (phi in the av3 annotation) was randomized. The full set of subtomogram positions was split into two independent half sets, one for all odd and one for all even subtomogram numbers. The two half sets were processed independently. For initial alignment and averaging, subtomograms with a box size of $64 \times 64 \times 64$ voxels (~$76.4 \times 76.4 \times 76.4$ nm$^3$) were extracted from the respective tomograms and averaged. The initial average shows a long, thin stalk.

After one iteration of alignment using a low pass filter of 30 Å and a complete in-plane angular search, twofold symmetry was applied for further iterations. During all steps of alignment and averaging, the missing wedge was modelled as the sum of the amplitude spectra determined from 100 randomly seeded noise positions for each tomogram. After six iterations, subtomograms with a box size of 96 × 96 × 96 voxels (~53.7 × 53.7 × 53.7 nm³) were re-extracted from tomograms with a voxel size of 5.97 Å. Subtomogram positions were reprojected onto the respective tomogram for visual inspection, and misaligned subtomogram positions as well as subtomograms that had converged to the same position were removed, reducing the final number of subtomograms to 2368 subtomograms (odd = 1182 and even = 1186). The two averages were then further refined for 4 iterations using a cylindrical mask tightly fit around the obtained structure and including the membrane and subsequently for six more iterations using a cylindrical mask excluding the membrane. To evaluate the final average and assess the resolution obtained, the averages from the two half sets were aligned to each other and the FSC was calculated as described in *Chen et al., 2013* indicating a final resolution of 25 Å (*Figure 4—figure supplement 2A*). To test if CTF correction of the data would improve the reconstruction obtained from subtomogram averaging, the defocus for each tilt image was determined using CTFFIND4 including phase shifts and the defocus estimate was visually checked against the power spectrum of the tilts. Using the determined defocused values CTF-corrected tomograms were generated in NovaCTF. Subtomograms were then re-extracted from Bin2 CTF-corrected tomograms, averaged, and aligned. Neither resolution nor map quality improved over the non CTF corrected data and, therefore, ultimately the map from the non-CTF corrected data was used as the final map.

The previously reported atomic coordinates of an AMPA subtype ionotropic glutamate receptor (*Sobolevsky et al., 2009*) (PDB: 3KG2) and NMDA receptor (PDB:5fxh) (*Tajima et al., 2016*) were fitted as a rigid body into the final subtomogram averaging map using the fit in map tool in UCSF Chimera. The former was a better fit in keeping with the expected higher abundance of AMPA receptors at the synapse (*Lowenthal et al., 2015*; *Spruston et al., 1995*).

## Cluster analysis

Clustering analysis of glutamate receptors was performed using the DBSCAN algorithm (minimal cluster size 4 and maximal distance between receptors of 70 nm) where the distances between pairs of receptors were defined as the shortest distance between their projections on the surface of the membrane. The mesh associated to the surface was estimated using the 'MyOpenCrust' implementation of the crust meshing method and simplified using the reducepatch function from Matlab. The distances between the projected coordinates were computed using the Dijkstra algorithm on the graph associated to the mesh.

We computationally categorised receptor clusters that were situated inside the cleft, in perisynaptic locations outside the cleft, or spanning the boundary of the synaptic cleft. If perisynaptic clusters arose because of detachment of the PoSM and PreSM during sample preparation, we would expect to find at least a small population of Psd95-GFP containing postsynaptic compartments that were not adhered to a pre-synaptic membrane. Examining every single Psd95-GFP puncta in our cryoCLEM dataset of 93 synapses marked a postsynaptic membrane that was also attached to a presynaptic membrane, suggesting synaptic contacts are established in the brain with a high avidity that is not affected by sample preparation.

## Acknowledgements

We would like to thank Christos Savvas, Giuseppe Cannone, and Shaoxia Chen for help with maintaining and setting up MRC LMB Titan Krios microscopes. We would like to thank the Astbury Biostructure Laboratory (ABSL), particularly Rebecca Thompson, Emma Hesketh, Dan Maskell, and Helmut Gnaegi for help with setting up and maintaining HPF, cryo-ultramicrotome, cryoFM, and Titan Krios microscopes. We are grateful to Jake Grimmett, Toby Darling, Wallace Tudeme, Kerry Kiggins, and Jack Carberry for providing essential computational support. Heather Lloyd, Ilona Rigo, Mel Reay, Danielle Mansfield, and Biomedical staff for technical support. Madeleine Gilbert and Eva Martínez Barceló for assistance with annotation of synapse tomograms. Grant funding: An Academy of Medical Sciences Springboard Award (SBF005/1046), UKRI Future Leader Fellowship (MR/V022644/1) and a University of Leeds Academic Fellowship to RAWF. CL was funded by a BBSRC White Rose DTP PhD studentship (BB/M011151/1). A Wellcome Trust (Technology Development Grant 202932), the

European Research Council (ERC) under the European Union's Horizon 2020 Research and Innovation Programme (695568 SYNNOVATE), Simons Foundation for Autism Research Initiative (529085) to SGNG and NHK. Medical Research Council as part United Kingdom Research and Innovation (MC_UP_1201/16) and the European Research Council (ERC) under the European Union's Horizon 2020 research and innovation programme (ERC-CoG-648432 MEMBRANEFUSION), and the Max Planck Society to JAGB. WK was supported by the Medical Research Council, as part of United Kingdom Research and Innovation (MC_UP_1201/8). ABSL Titan Krios microscopes were funded by the University of Leeds and Wellcome Trust (108466/Z/15/Z). The Leica EM ICE, UC7 ultra/cryo-ultramicrotome and cryoCLEM systems were funded by Wellcome Trust (208395/Z/17/Z).

## Additional information

### Funding

| Funder | Grant reference number | Author |
|---|---|---|
| Academy of Medical Sciences | SBF005/1046 | René AW Frank |
| UK Research and Innovation | MR/V022644/1 | René AW Frank |
| Biotechnology and Biological Sciences Research Council | BB/M011151/1 | Charlie Lovatt |
| Wellcome Trust | 10.35802/202932 | Seth GN Grant |
| European Research Council | 10.3030/695568 | Seth GN Grant |
| Simons Foundation | 529085 | Noboru H Komiyama Seth GN Grant |
| Medical Research Council | MC_UP_1201/16 | John AG Briggs |
| European Research Council | 10.3030/648432 | John AG Briggs |
| Medical Research Council | MC_UP_1201/8 | Wanda Kukulski |

The funders had no role in study design, data collection and interpretation, or the decision to submit the work for publication. For the purpose of Open Access, the authors have applied a CC BY public copyright license to any Author Accepted Manuscript version arising from this submission.

### Author contributions

Julia Peukes, Data curation, Software, Formal analysis, Investigation, Methodology, Writing – review and editing; Charlie Lovatt, Formal analysis, Investigation, Visualization, Writing – review and editing; Conny Leistner, Investigation, Methodology, Writing – review and editing; Jerome Boulanger, Software, Formal analysis, Investigation, Writing – review and editing; Dustin R Morado, Formal analysis, Investigation, Methodology, Writing – review and editing; Martin JG Fuller, Investigation, Writing – review and editing; Wanda Kukulski, Resources, Methodology, Writing – review and editing; Fei Zhu, Noboru H Komiyama, Resources, Investigation, Writing – review and editing; John AG Briggs, Software, Supervision, Methodology, Writing – review and editing; Seth GN Grant, Resources, Supervision, Methodology, Writing – review and editing; René AW Frank, Conceptualization, Data curation, Formal analysis, Supervision, Funding acquisition, Investigation, Visualization, Methodology, Writing – original draft, Project administration, Writing – review and editing

### Author ORCIDs

Julia Peukes ![ORCID] https://orcid.org/0000-0001-8560-7407
Charlie Lovatt ![ORCID] https://orcid.org/0000-0002-7171-0494
Jerome Boulanger ![ORCID] https://orcid.org/0000-0003-0237-3743
Wanda Kukulski ![ORCID] https://orcid.org/0000-0002-2778-3936

John AG Briggs https://orcid.org/0000-0003-3990-6910
Seth GN Grant https://orcid.org/0000-0001-8732-8735
René AW Frank https://orcid.org/0000-0001-9724-9547

### Ethics

Animals were treated in accordance with UK Animal Scientific Procedures Act (1986) and NIH and ARRIVE guidelines. All animal experiments were approved by theUniversity of Leeds Animal Welfare and Ethics Committee.

Reviewer #1 (Public review): https://doi.org/10.7554/eLife.100335.3.sa1
Reviewer #2 (Public review): https://doi.org/10.7554/eLife.100335.3.sa2
Reviewer #3 (Public review): https://doi.org/10.7554/eLife.100335.3.sa3
Author response https://doi.org/10.7554/eLife.100335.3.sa4

## Additional files

### Supplementary files
MDAR checklist

### Data availability

Subtomogram average maps and a representative tomogram have been deposited in the Electron Microscopy Data Bank (EMDB) under accession codes EMD-54246 and EMD-53528, respectively. Tilt series and 4x binned tomographic volumes of 'ultra-fresh' and in-tissue synapses (Figure 1-table 1 and Figure 2-table 1) and segmentation coordinates have been deposited in the Electron Microscopy Public Image Archive (EMPIAR) with the accession code EMPIAR-12795.

The following datasets were generated:

| Author(s) | Year | Dataset title | Dataset URL | Database and Identifier |
|---|---|---|---|---|
| Peukes J, Briggs JAG, Frank RAW | 2025 | Subtomogram average maps and a representative tomogram | https://www.emdataresource.org/EMD-54246 | EMDataResource, EMD-54246 |
| Frank RAW, Peukes J, Briggs JAG | 2025 | Subtomogram average maps and a representative tomogram | https://www.emdataresource.org/EMD-53528 | EMDataResource, EMD-53528 |
| Lovatt C, Frank RAW | 2025 | cryoCLEM-guided cryoET datasets | https://www.ebi.ac.uk/empiar/EMPIAR-12795 | Electron Microscopy Public Image Archive, EMPIAR-12795 |

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
