## [Editor Report · eLife Assessment]

Peukes et al. report **compelling** ultrastructures of excitatory synapses in the mouse forebrain that will serve as a reference for future work in the field. Their **important** findings using correlated fluorescence and cryo-electron tomography challenge the textbook view of synaptic structure that emerged from chemically fixed and metal-stained tissues. Instead of a post-synaptic density, these authors reveal the architecture of the cytoskeletal, neurotransmitter receptor clusters, and organelles in the 'synaptoplasm'.

---

## [Referee Report · Reviewer #1 (Public review)]

The authors survey the ultrastructural organization of glutamatergic synapses by cryo-ET and image processing tools using two complementary experimental approaches. The first approach employs so-called "ultra-fresh" preparations of brain homogenates from a knock-in mouse expressing a GFP-tagged version of PSD-95, allowing Peukes and colleagues to specifically target excitatory glutamatergic synapses. In the second approach, direct in-tissue (using cortical and hippocampal regions) targeting of the glutamatergic synapses employing the same mouse model is presented. In order to ascertain whether the isolation procedure causes any significant changes in the ultrastructural organization (and possibly synaptic macromolecular organization) the authors compare their findings using both of these approaches. The quantitation of the synaptic cleft height reveals an unexpected variability, while the STA analysis of the ionotropic receptors provides insights into their distribution with respect to the synaptic cleft.

The main novelty of this study lies in the continuous claims by the authors that the sample preservation methods developed here are superior to any others previously used. This leads them as well to systematically downplay or directly ignore a substantial body of previous cryo-ET studies of synaptic structure. Without comparisons with the cryo-ET literature, it is very hard to judge the impact of this work in the field. Furthermore, the data does not show any better preservation in the so-called "ultra-fresh" preparation than in the literature, perhaps to the contrary as synapses with strangely elongated vesicles are often seen. Such synapses have been regularly discarded for further analysis in previous synaptosome studies (e.g. Martinez-Sanchez 2021). Whilst the targeting approach using a fluorescent PSD95 marker is novel and seems sufficiently precise, the authors use a somewhat outdated approach (cryo-sectioning) to generate in-tissue tomograms of poor quality. To what extent such tomograms can be interpreted in molecular terms is highly questionable. The authors also don't discuss the physiological influence of 20% dextran used for high-pressure freezing of these "very native" specimens.

Lastly, a large part of the paper is devoted to image analysis of the PSD which is not convincing (including a somewhat forced comparison with the fixed and heavy-metal staining room temperature approach). Despite being a technically challenging study, the results fall short of expectations.

---

## [Referee Report · Reviewer #2 (Public review)]

Summary:

The authors set out to visualize the molecular architecture of the adult forebrain glutamatergic synapses in a near-native state. To this end, they use a rapid workflow to extract and plunge-freeze mouse synapses for cryo-electron tomography. In addition, the authors use knockin mice expression PSD95-GFP in order to perform correlated light and electron microscopy to clearly identify pre- and synaptic membranes. By thorough quantification of tomograms from plunge- and high-pressure frozen samples, the authors show that the previously reported 'post-synaptic density' does not occur at high frequency and therefore not a defining feature of a glutamatergic synapse.

Subsequently, the authors are able to reproduce the frequency of post-synaptic density when preparing conventional electron microscopy samples, thus indicating that density prevalence is an artifact of sample preparation. The authors go on to describe the arrangement of cytoskeletal components, membraneous compartments, and ionotropic receptor clusters across synapses.

Demonstrating that the frequency of the post-synaptic density in prior work is likely an artifact and not a defining feature of glutamatergic synapses is significant. The descriptions of distributions and morphologies of proteins and membranes in this work may serve as a basis for the future of investigation for readers interested in these features.

Strengths:

The authors perform a rigorous quantification of the molecular density profiles across synapses to determine the frequency of the post-synaptic density. They prepare samples using two cryogenic electron microscopy sample preparation methods, as well as one set of samples using conventional electron microscopy methods. The authors can reproduce previous reports of the frequency of the post-synaptic density by conventional sample preparation, but not by either of the cryogenic methods, thus strongly supporting their claim.

---

## [Referee Report · Reviewer #3 (Public review)]

Summary:

The authors use cryo-electron tomography to thoroughly investigate the complexity of purified, excitatory synapses. They make several major interesting discoveries: polyhedral vesicles that have not been observed before in neurons; analysis of the intermembrane distance, and a link to potentiation, essentially updating distances reported from plastic-embedded specimen; and find that the postsynaptic density does not appear as a dense accumulation of proteins in all vitrified samples (less than half), a feature which served as a hallmark feature to identify excitatory plastic-embedded synapses.

Strengths:

(1) The presented work is thorough: the authors compare purified, endogenously labeled synapses to wild-type synapses to exclude artifacts that could arise through the homogenation step, and, in addition, analyse plastic embedded, stained synapses prepared using the same quick workflow, to ensure their findings have not been caused by way of purification of the synapses. Interestingly, the 'thick lines of PSD' are evident in most of their stained synapses.

(2) I commend the authors on the exceptional technical achievement of preparing frozen specimens from a mouse within two minutes.

(3) The approaches highlighted here can be used in other fields studying cell-cell junctions.

(4) The tomograms will be deposited upon publication which will enable neurobiologists and researchers from other fields to carry on data evaluation in their field of expertise since tomography is still a specialized skill and they collected and reconstructed over 100 excellent tomograms of synapses, which generates a wealth of information to be also used in future studies.

(5) The authors have identified ionotropic receptor positions and that they are linked to actin filaments, and appear to be associated with membrane and other cytosolic scaffolds, which is highly exciting.

(6) The authors achieved their aims to study neuronal excitatory synapses in great detail, were thorough in their experiments, and made multiple fascinating discoveries. They challenge dogmas that have been in place for decades and highlight the benefit of implementing and developing new methods to carefully understand the underlying molecular machines of synapses.

Impact on community:

The findings presented by Peukes et al. pertaining to synapse biology change dogmas about the fundamental understanding of synaptic ultrastructure. The work presented by the authors, particularly the associated change of intermembrane distance with potentiation and the distinct appearance of the PSD as an irregular amorphous 'cloud' will provide food for thought and an incentive for more analysis and additional studies, as will the discovery of large membranous and cytosolic protein complexes linked to ionotropic receptors within and outside of the synaptic cleft, which are ripe for investigation. The findings and tomograms available will carry far in the synapse fields and the approach and methods will move other fields outside of neurobiology forward. The method and impactful results of preparing cryogenic, unlabeled, unstained, near-native synapses may enable the study of how synapses function at high resolution in the future.

---

## [Author Response]

The following is the authors’ response to the original reviews.

**Reviewer #1 (Public review):**
The authors survey the ultrastructural organization of glutamatergic synapses by cryo-ET and image processing tools using two complementary experimental approaches. The first approach employs so-called "ultra-fresh" preparations of brain homogenates from a knock-in mouse expressing a GFP-tagged version of PSD-95, allowing Peukes and colleagues to specifically target excitatory glutamatergic synapses. In the second approach, direct in-tissue (using cortical and hippocampal regions) targeting of the glutamatergic synapses employing the same mouse model is presented. In order to ascertain whether the isolation procedure causes any significant changes in the ultrastructural organization (and possibly synaptic macromolecular organization) the authors compare their findings using both of these approaches. The quantitation of the synaptic cleft height reveals an unexpected variability, while the STA analysis of the ionotropic receptors provides insights into their distribution with respect to the synaptic cleft.The main novelty of this study lies in the continuous claims by the authors that the sample preservation methods developed here are superior to any others previously used. This leads them as well to systematically downplay or directly ignore a substantial body of previous cryo-ET studies of synaptic structure. Without comparisons with the cryo-ET literature, it is very hard to judge the impact of this work in the field. Furthermore, the data does not show any better preservation in the so-called "ultra-fresh" preparation than in the literature, perhaps to the contrary as synapses with strangely elongated vesicles are often seen. Such synapses have been regularly discarded for further analysis in previous synaptosome studies (e.g. Martinez-Sanchez 2021). Whilst the targeting approach using a fluorescent PSD95 marker is novel and seems sufficiently precise, the authors use a somewhat outdated approach (cryo-sectioning) to generate in-tissue tomograms of poor quality. To what extent such tomograms can be interpreted in molecular terms is highly questionable. The authors also don't discuss the physiological influence of 20% dextran used for high-pressure freezing of these "very native" specimens.Lastly, a large part of the paper is devoted to image analysis of the PSD which is not convincing (including a somewhat forced comparison with the fixed and heavy-metal staining room temperature approach). Despite being a technically challenging study, the results fall short of expectations.

Our manuscript contains a discussion of both conventional EM and cryoET of synapses. We apologise if we have omitted referencing or discussing any earlier cryoET work. This was certainly not our intention, and we include a more complete discussion of published cryoET work on synapses in our revised manuscript.

The reviewer is concerned that the synaptic vesicles in some synapse tomograms are “stretched” and that this may reflect poor preservation. We would like to point out that such non-spherical synaptic vesicles have also been previously reported in cryoET of primary neurons grown on EM grids (Tao et al., J. Neuro, 2018). Indeed, there is no reason per se to suppose synaptic vesicles are always spherical and there are many diverse families of proteins expressed at the synapse that shape membrane curvature (BAR domain proteins, synaptotagmin, epsins, endophilins and others). We will add further discussion of this issue in the revised manuscript.

The reviewer regards ‘cryo-sectioning’ as outdated and cryoET data from these preparations as “poor quality”. We respectfully disagree. Preparing brain tissues for cryoET is generally considered to be challenging. The first successful demonstration of preparing such samples was before the advent of the cryoEM resolution revolution (with electron counting detectors) by Zuber et al (Proc. Natl. Acad. Sci.,2005) preparing cryo-sections/CEMOVIS of in vitro brain cultures. We followed this technique to prepare tissue cryo-sections for cryoET in our manuscript. Recently, cryoFIB-SEM liftout has been developed as an alternative method to prepare tissue samples for cryoET (Mahamid et al., J. Struct. Biol., 2015) and only more recently this method became available to more laboratories. Both techniques introduce damage as has been described (Han et al., J. Microsc., 2008; Lucas et al., Proc. Natl. Acad. Sci., 2023). Importantly no like-for-like, quantitative comparison of these two methodologies has yet been performed. We have recently demonstrated that the molecular structure of amyloid fibrils within human brain is preserved down to the protein fold level in samples prepared by cryo-sectioning (Gilbert et al., Nature, 2024). We will add further detail on the process by which we excluded poor quality tomograms from our analysis, which we described in detail in our methods section.

The reviewer asks what the physiological effect is of adding 20% w/v ~40,000 Da dextran? This is a reasonable concern since this could in principle exert osmotic pressure on the tissue sample. While we did not investigate this ourselves, earlier studies have (Zuber *et al*, 2005) showing cell membranes were not damaged by and did not have any detectable effect on cell structure in the presence of this concentration of dextran.

The reviewer is not convinced by our analysis of the apparent molecular density of macromolecules in the postsynaptic compartment that in conventional EM is called the postsynaptic density. However, the reviewer provides no reasoning for this assessment nor alternative approaches that could be attempted. We would like to add that we have tested multiple different approaches to objectively measure molecular crowding in cryoET data, that give comparable results. We believe that our conclusion – that we do not observe an increased molecular density conserved at the postsynaptic membrane, and that the PSD that we and others observed by conventional EM does not correspond to a region of increased molecular density - is well supported by our data. We and the other reviewers consider this an important and novel observation.

**Reviewer #2 (Public review)**:Summary:The authors set out to visualize the molecular architecture of the adult forebrain glutamatergic synapses in a near-native state. To this end, they use a rapid workflow to extract and plunge-freeze mouse synapses for cryo-electron tomography. In addition, the authors use knockin mice expression PSD95-GFP in order to perform correlated light and electron microscopy to clearly identify pre- and synaptic membranes. By thorough quantification of tomograms from plunge- and high-pressure frozen samples, the authors show that the previously reported 'post-synaptic density' does not occur at high frequency and therefore not a defining feature of a glutamatergic synapse.Subsequently, the authors are able to reproduce the frequency of post-synaptic density when preparing conventional electron microscopy samples, thus indicating that density prevalence is an artifact of sample preparation. The authors go on to describe the arrangement of cytoskeletal components, membraneous compartments, and ionotropic receptor clusters across synapses.Demonstrating that the frequency of the post-synaptic density in prior work is likely an artifact and not a defining feature of glutamatergic synapses is significant. The descriptions of distributions and morphologies of proteins and membranes in this work may serve as a basis for the future of investigation for readers interested in these features.Strengths:The authors perform a rigorous quantification of the molecular density profiles across synapses to determine the frequency of the post-synaptic density. They prepare samples using two cryogenic electron microscopy sample preparation methods, as well as one set of samples using conventional electron microscopy methods. The authors can reproduce previous reports of the frequency of the post-synaptic density by conventional sample preparation, but not by either of the cryogenic methods, thus strongly supporting their claim.

We thank the reviewer for their generous assessment of our manuscript.

**Reviewer #3 (Public review):**
Summary:The authors use cryo-electron tomography to thoroughly investigate the complexity of purified, excitatory synapses. They make several major interesting discoveries: polyhedral vesicles that have not been observed before in neurons; analysis of the intermembrane distance, and a link to potentiation, essentially updating distances reported from plastic-embedded specimen; and find that the postsynaptic density does not appear as a dense accumulation of proteins in all vitrified samples (less than half), a feature which served as a hallmark feature to identify excitatory plastic-embedded synapses.Strengths:(1)The presented work is thorough: the authors compare purified, endogenously labeled synapses to wild-type synapses to exclude artifacts that could arise through the homogenation step, and, in addition, analyse plastic embedded, stained synapses prepared using the same quick workflow, to ensure their findings have not been caused by way of purification of the synapses. Interestingly, the 'thick lines of PSD' are evident in most of their stained synapses.(2)I commend the authors on the exceptional technical achievement of preparing frozen specimens from a mouse within two minutes.(3)The approaches highlighted here can be used in other fields studying cell-cell junctions.(4)The tomograms will be deposited upon publication which will enable neurobiologists and researchers from other fields to carry on data evaluation in their field of expertise since tomography is still a specialized skill and they collected and reconstructed over 100 excellent tomograms of synapses, which generates a wealth of information to be also used in future studies.(5) The authors have identified ionotropic receptor positions and that they are linked to actin filaments, and appear to be associated with membrane and other cytosolic scaffolds, which is highly exciting.(6) The authors achieved their aims to study neuronal excitatory synapses in great detail, were thorough in their experiments, and made multiple fascinating discoveries. They challenge dogmas that have been in place for decades and highlight the benefit of implementing and developing new methods to carefully understand the underlying molecular machines of synapses.Weaknesses:The authors show informative segmentations in their figures but none have been overlayed with any of the tomograms in the submitted videos. It would be helpful for data evaluation to a broad audience to be able to view these together as videos to study these tomograms and extract more information. Deposition of segmentations associated with the tomgrams would be tremendously helpful to Neurobiologists, cryo-ET method developers, and others to push the boundaries.Impact on community:The findings presented by Peukes et al. pertaining to synapse biology change dogmas about the fundamental understanding of synaptic ultrastructure. The work presented by the authors, particularly the associated change of intermembrane distance with potentiation and the distinct appearance of the PSD as an irregular amorphous 'cloud' will provide food for thought and an incentive for more analysis and additional studies, as will the discovery of large membranous and cytosolic protein complexes linked to ionotropic receptors within and outside of the synaptic cleft, which are ripe for investigation. The findings and tomograms available will carry far in the synapse fields and the approach and methods will move other fields outside of neurobiology forward. The method and impactful results of preparing cryogenic, unlabelled, unstained, near-native synapses may enable the study of how synapses function at high resolution in the future.

We thank the reviewer for their supportive assessment of our manuscript. We thank the reviewer for suggesting overlaying segmentations with videos of the raw tomographic volumes. We will include this in our revised manuscript.

**Reviewer #1 (Recommendations for the authors):**
Major comments:(1) The previous literature on synaptic cryo-ET studies is systematically ignored. The results presented here (and their novelty) must be compared directly with this body of work, rather than with classical EM.

Our submitted manuscript included a 3-paragraph discussion of earlier synaptic cryoET studies, albeit we apologize that a seminal citation was missing, which we have corrected in our revised manuscript. We have now also included an additional brief discussion related to several more recent cryoET studies (see citations below) that were published after our pre-print was first deposited in 2021.

(1) Held, R.G., Liang, J., and Brunger, A.T. (2024). Nanoscale architecture of synaptic vesicles and scaffolding complexes revealed by cryo-electron tomography. Proc. Natl. Acad. Sci. *121*, e2403136121. https://doi.org/10.1073/pnas.2403136121.

(2) Held, R.G., Liang, J., Esquivies, L., Khan, Y.A., Wang, C., Azubel, M., and Brunger, A.T. (2024). In-Situ Structure and Topography of AMPA Receptor Scaffolding Complexes Visualized by CryoET. bioRxiv, 2024.10.19.619226. https://doi.org/10.1101/2024.10.19.619226.

(3)Matsui, A., Spangler, C., Elferich, J., Shiozaki, M., Jean, N., Zhao, X., Qin, M., Zhong, H., Yu, Z., and Gouaux, E. (2024). Cryo-electron tomographic investigation of native hippocampal glutamatergic synapses. eLife *13*, RP98458. https://doi.org/10.7554/elife.98458.

(4)Glynn, C., Smith, J.L.R., Case, M., Csöndör, R., Katsini, A., Sanita, M.E., Glen, T.S., Pennington, A., and Grange, M. (2024). Charting the molecular landscape of neuronal organisation within the hippocampus using cryo electron tomography. bioRxiv, 2024.10.14.617844. https://doi.org/10.1101/2024.10.14.617844.

We discuss the above papers in our revised manuscript with the following:

“Since submission of our manuscript, several reports of synapse cryoET from within cultured primary neurons (Held et al., 2024a, 2024b) and mouse brain(Glynn et al., 2024; Matsui et al., 2024) were prepared by cryoFIB-milling. These new datasets are largely consistent with the data reported here. CryoFIB-SEM has the advantage of overcoming the local knife damage caused by cryo-sectioning but introduces amorphization across the whole sample that diminishes the information content (Al-Amoudi et al., 2005; Lovatt et al., 2022; Lucas and Grigorieff, 2023). We have recently shown cryoET data is capable of revealing subnanometer resolution in-tissue protein structure from vitreous cryo-sections (Gilbert et al., 2024) and near-atomic structures within cryo-sections has recently been demonstrated (Elferich et al., 2025).”

Although there is variation between individual synapses, PSDs are clearly visible in several previous cryo-ET studies (even if it's not as striking as in heavy-metal stained samples). In fact, although the contrast of the images is generally poor, PSDs are also visible in several examples shown in Figure 1 - Supplement 3. Not being able to detect them seems more of a problem of the workflow used here than of missing features. The authors should also discuss why heavy-metal stains would accumulate on a non-existing structure (PSD) in conventional EM.

We agree that apparent higher molecular density can be observed in example tomographic data of earlier cryoET studies. We also report individual examples of similar synapses in our dataset. A key strength of our approach is that we have assessed the molecular architecture of large numbers of adult brain synapses acquired by an unbiased approach (solely guided by PSD95 cryoCLEM), which indicate that a higher molecular density proximal to the postsynaptic membrane is not a conserved feature of glutamatergic synapses in the adult brain. There is no rationale for our cryoCLEM approach being a ‘problem of the workflow’.

The reviewer misunderstands the weaknesses of conventional/room temperature EM workflows (including resin-embedding and freeze substitution). It is unavoidable that most proteins are damaged by denaturation and/or washed away by washing samples in organic solvents (methanol/acetone that directly denature most proteins) during tissue preparation for conventional EM. It is therefore conceivable that in such preparations a relative increase in contrast proximal to the postsynaptic membrane (‘PSD’) would appear if cytoplasmic proteins were washed away during these harsh organic solved washing steps, leaving only those denatured proteins that are tethered to the postsynaptic membrane. It is not that the PSD is absent in cryoEM, rather that this difference in molecular crowding is not evident when tissues are imaged directly by cryoEM and have not undergone the harsh sample preparation required for conventional/room temperature EM.

(2) Whether the synapses examined here are in a more physiological state than those analyzed in other papers remains absolutely unclear. For example, the quality of the tomographic slice shown in Figure 1C is poor, with the majority of synaptic vesicles looking suspiciously elongated.

We addressed this in our public reviews.

(3) How were actin filaments segmented and quantified (e.g. for Fig 1E)? Apart from actin, can the authors show some examples of other macromolecular complexes (e.g. ribosomes) that they are able to identify in synapses (based on the info in supplementary tables)? Also, the mapping of glutamatergic receptors is not convincing, as the molecules were picked manually. To analyze their distribution, they should be mapped as comprehensively as possible by e.g. template matching.

Actin filaments identified by ~7 nm diameter with ~70° branch points were manually segmented in IMOD. The number of filaments was counted per postsynaptic compartment. We have amended the methods section to include this description.

“In the PoSM, F-actin formed a network with ~70° branch points (Figure 1–figure supplement 1C) likely formed by Arp2/3, as expected(Pizarro-Cerdá 2017,Fäßler 2020) . Putative filament copy number in the PoSM was estimated by manual segmentation in IMOD.” Manual picking was validated by the quality of the subtomogram average, which although only reached modest resolution (25 Å) is consistent with the identification of ionotropic glutamate receptors.

(4) In the section "Synaptic organelles" the authors should provide some general information on the average number and size of synaptic vesicles (for the in-tissue tomograms).

We have provided this information in the methods section:

“The average diameter of synaptic vesicles was 40.2 nm and the minimum and maximum dimensions ranged from 20 to 57.8 nm, measured from the outside of the vesicle that included ellipsoidal synaptic vesicles similar to those previously reported (Tao et al., 2018).” A detailed survey of the presynaptic compartment, including the number of presynaptic vesicles was not the focus of our manuscript. We have deposited all tomograms from our dataset for any further data mining.

Can the "flat tubular membranes compartments" be attributed to ER? The angular vesicles certainly have a typical ER appearance, as such morphology has been seen in several cryo-ET studies of neuronal and non-neuronal cells.

In neuronal cells we regard it as unsafe to describe an intracellular organelle as being endoplasmic reticulum on the basis of morphology alone (eg. Smooth ER described widely in conventional EM) because of the apparent diversity of distinct organelles. As described in our methods section, we could have confidence that a membrane compartment is ER when we observe ribosomes tethered to the membrane. In instances where flat/tubular membranes did not have associated ribosomes, we take the cautious view that there is not sufficient evidence to define these as ER.

Importantly, polyhedral vesicles were distinct from the flat/tubular membranes that resembled ER and are at present organelles of unknown identity. It will be important in future experiments to determine what are the protein constituents of these distinct organelle types to understand both their functions and how these distinct membrane architectures are assembled.

Therefore, the sentences in lines 198-199 are simply wrong. Additionally, features of even higher membrane curvature are common in the ER (e.g. Collado et al., Dev Cell 2019).

We thank the reviewer for bringing our attention to this excellent paper (Collado et al.). We agree that the sentence describing the curvature being higher than all other membranes except mitochondrial cristae is wrong. We have removed this sentence in the revised manuscript.

(5)The quality of the tomographic data for the in-tissue sample is low, likely due to cryo-sectioning-induced artifacts, as extensively documented in the literature. Additionally, the authors used 20% dextran as cryo-protectant for high-pressure freezing, which contrasts with statements like those in lines 342-344. Given that several publications describing the in-tissue targeting of synapses (e.g. from Eric Gouaux's lab) are available, the quality of the tomographic data presented in this work is underwhelming and limits the conclusions that can be drawn, not providing a solid basis for future studies of in-tissue synapse targeting. However, the complete workflow (excluding the sectioning part) can be adapted for a cryo-FIB approach. The authors should discuss the limitations of their approach.

Our manuscript preprint was deposited in the Biorxiv several years before Matsui/Gouaux’s recent ELife paper that reported a novel work-flow for in-tissue cryoET. It is difficult to directly compare data from our and Matsui/Gouaux’s approach because the latter reported a dataset of only 3 tomograms. Note also that Matsui/Gouaux followed our approach of using 20% dextran 40,000 as a cryo-preservative. The use of 20% dextran 40,000 as a cryo-protectant was first established by Zuber et al., 2005 (PMID: 16354833) and shown avoid hyper-osmotic pressure and cell membrane rupture. However, Matsui/Gouaux additionally included 5% sucrose in their cryoprotectant. We did not include sucrose as cryo-preservative because this exerts osmotic pressure and was not necessary to achieve vitreous tissues in our workflow.

Before high-pressure freezing, Matsui/Gouaux also incubated tissue slices in a HEPES-buffered artificial cerebrospinal fluid (that included 2 mM CaCl2 but did not include glucose as an energy source) for 1 h at room temperature to label AMPA receptors with Fab fragment-Au conjugates. Under these conditions, neurons can elicit both physiological and excitotoxic action potentials (even though AMPARs were themselves antagonised with ZK-200775). The absence of glucose is a concern, and it is unclear to what extent tissue viability is affected by this incubation step. In contrast, we chose to use an NMDG-based artificial cerebrospinal fluid for slice preparation and high-pressure freezing that is a well-established method for preserving neuronal viability (Ting et al., 2018).

We addressed the supposed limitations of cryo-sectioning versus cryoFIB-SEM in our public response. In particular, we have recently shown that cryo-sectioning produced a subnanometer resolution in-tissue structure of a protein, that has so far only been achieved for ribosome within cryoFIB-SEM sample preparations. A discussion of cryo-sectioning versus cryoFIB-SEM must be informed by new data that directly compares these methods, which is not the subject of our eLife paper. We also cite a recent preprint directly comparing cryoFIB-milled lamellae with cryo-sections and showing that near atomic resolution structures can also be obtained from the latter sample preparations (Elferich et al., 2025).

(6) The authors show (in Supplementary) putative tethers connecting SV and the plasma membrane. Is it possible to improve the image quality (e.g. some sort of filtering or denoising) so that the tethers appear more obvious? Can the authors observe connectors linking synaptic vesicles?

We have tested multiple iterative reconstruction and denoising approaches, including SIRT and noise2noise filtering in Isonet. We observed instances of macromolecular complexes linking one synaptic vesicle with another. However, there was no question we sought to answer by performing a quantitative analysis of these linkers.

(7) Figure 4F is missing.

Thank you for spotting this omission. We have corrected this in the revised manuscript.

(8) Most quantifications lack statistical analyses. These need to be included, and only statistically significant findings should be discussed. Terms like "significantly" (e.g. Line 144) should only be used in these cases.

We used the term ‘significantly’ in the results section line 143 and line 166 in revised text, we cite figure 1H and 2F showing analyses in which we have in fact performed statistical tests (t-tests with Bonferroni correction) comparing the voxel intensities in regions of the cytoplasm that are proximal versus distal to the postsynaptic membrane. We have amended the main text to include the details of the statistical test that we performed. Also, we neglected to include a description of the statistical test in line 241, which cites Figure 3G. We have corrected this in the revised text.

Minor comments:(1) Can the authors comment on why only 1-2 grids are prepared per mouse brain (in M&M -section)?

We prepared only two grids in order to have prepared samples within 2 minutes, to limit deterioration of the sample.

(2) Figure 1 Supplement 2 and its legend are confusing (averaging of non-aligned versus aligned post-synaptic membrane). Can the authors describe more clearly their molecular density profile analysis?

We apologise that this figure legend was insufficient. We have included a detailed description of our molecular density profile analysis in the methods section entitled ‘Molecular density profile analysis’. In the revised manuscript we have now also included a citation to this methods section in Figure – figure 1 supplement 2 legend.

(3) Please clarify with higher precision the areas were recorded in relation to the fluorescent spots (e.g. Figures 3A-C).

We have included a white rectangular annotation in the cryoCLEM inset panels of Figures 3A-C to indicate the field of view of each corresponding tomographic slice. This shows that PSD95-GFP puncta localise to the postsynaptic compartments in each tomogram.

(4) Figure 4 Supplement 2D is not clear: the connection between receptors and actin should be shown in a segmentation.

We agree with the reviewer. A ‘connection’ is not clear, which is expected because the cytoplasmic domain of ionotropic glutamate receptor subunits is composed of a non-globular/intrinsically disordered sequence. We have amended our description of the proximity of actin cytoskeleton to ionotropic glutamate receptor clusters in the main text replacing “associated with” to “adjacent to”.

(5) Line 341: the reference is referred to by a number (56) at the end of the sentence, rather than by name.

Good spot. We have corrected this in the revised manuscript.

(6) Line 968: tomograms is misspelled.

Good spot. We have corrected this error (line 1018 in our revised manuscript).

**Reviewer #2 (Recommendations for the authors):**
(1) On page 11: "The position of (i)onotropic receptor...".

Good spot. We have corrected this.

(2) On page 13: "Slightly higher relative molecular density..." this line ends with a citation to reference '56', but the works cited are not numbered.

Good spot. We have corrected this in the revised manuscript.

(3) On page 46: "as described in (69)..." the works cited are not numbered.

Good spot. We have corrected this in the revised manuscript.

**Reviewer #3 (Recommendations for the authors):**
(1) The title does not do the work justice. The authors make many exciting discoveries, e.g. PSD appearance, new polyhedral vesicles, ionotropic receptor positions, and intermembrane distance changes even within the synaptic cleft, but title their manuscript "The molecular infrastructure of glutamatergic synapses in the mammalian forebrain". It is also a bit misleading, since one would have expected more molecular detail and molecular maps as part of the work, so the authors may think about updating the title to reflect their exciting work.

We thank the reviewer for recognising the exciting discoveries in our manuscript. Summarising all these in a title is challenging. We intend ‘molecular infrastructure’ to mean a structure composed of many molecules including proteins (by analogy ‘transport infrastructure’ is composed of many roads, ports and train lines).

(2) It would be in the spirit of eLife and open science if the authors could submit their segmentations alongside the tomographic data to either EMPIAR or pdb-dev (if they accept it) or the new CZII cryoET data portal for neurobiologists, method developers, and others to use.

We agree with the reviewer. We have deposited in subtomogram averaged map of AMPA receptor in EMDB, and all tilt series and 4x binned tomographic reconstructions described in our manuscript (figure 1- table1 and figure 2 -table 2), together with segmentations in EMPIAR.

(3) Methods: the authors establish an exciting new workflow to get from living mice to frozen specimens within 2 minutes and perform many unique analyses that would be useful to different fields. Their methods section overall is well described and contains criteria and details that should allow others to apply experiments to their scientific problems. However, it would be very helpful to expand on the methods in the 'annotation and analysis [...]' and "Subtomogram averaging" sections, to at least in short describe the steps without having to embark on a reference journey for each method and generally provide more detail. For the annotation section, the software used for annotation is not listed. Table 1 only contains the list of the counts of organelles etc. identified in each tomogram, no processing details.

We have revised the methods section ‘annotation and analysis’ including software used (IMOD). We have also included a slightly more detailed description of subtomogram averaging. We did not include ‘processing details’ because there are none - identification of constituents in each tomogram was carried out manually, as described in the methods section.

(4) Some of the tomograms submitted as videos may have slipped through as an early version since they appear to be originating from not perfectly aligned tiltseries; vesicles and membranes can be observed 'rubberbanding'. The authors should go through and check their videos.

We thank the referee for suggesting we double check our tomogram videos. All movies are representative tomographic reconstructions from ultra-fresh synapse preparations (Figure 1 – videos 1-7) and synapses in tissue cryo-sections (Figure 2 – videos 1-2). We have double checked that the videos correspond to tomograms that were aligned as good as possible. In general, tissue cryo-section tomograms reconstructed less well than ultra-fresh synapse tomograms, which limits the information content of these data, as expected. Consequently, the reconstructions shown in these videos were all reconstructed as best we could (testing multiple approaches in IMOD, and more recent software packages, eg. AreTomo). While we think it is important to share all tomograms, regardless of quality, we were careful to exclude tomograms for analysis that did not contain sufficient information for analysis (as described in the methods section).

**Minor suggestions:**
(1) Page 13, line 341, reference 56, but references are not numbered. Please update.

Good spot. We have corrected this in the revised manuscript.

(2) Page 33, line 746, the figure legend is not referencing the correct figure panels G-K should be I-K;

We have amended the Figure 3 legend to “(G-K) Snapshots and quantification of membrane remodeling within glutamatergic synapses”.

(3) Page 33, line 750; reads 'same as E', but should be 'same as G'.

Good spot. We have corrected this in the revised manuscript.

(4) Page 35, Figure 4: Please use more labels: Figure 4B: it would be helpful to use different colors for each view and match to the tomogram - then non-experts could easily relate the projections and real data; Figure 4C: please label domains; Figure 4F: the figure panel got lost.

This is an interesting idea. While our subtomgram average of 2522 subvolumes provided decent evidence that these are ionotropic receptors, we are reluctant to label specific putative domains of individual subvolumes in the raw tomographic slice because the resolution of the raw tomogram (particularly in the Z-direction) is worse and may not be sufficient to resolve definitely each domain layer. We hope the reviewer appreciates our cautious approach.

(5) Page 42, line 933: incomplete sentence.

Good spot. We have corrected this in the revised manuscript.

(6) Page 46, line 1038; Reference 69 is in brackets, but references are not numbered. Please update.

Good spot. We have corrected this in the revised manuscript.